# The global, regional and national burden of pancreatitis due to alcohol use: Results from the Global Burden of Disease Study 2021 and projections to 2040

Letai Li[1,2�], Yaowen Zhang[1�], Jiajie Leng[1,2�], Siyu Li[1], Yang Lei[1], Zhenrui Cao[1,2], Yuxiang Luo[3], Haibing Xiong[4], Zhongjun Wu[5*], Rui Tao[6*], Yingjiu Jiang[1,2*]

**1** Chongqing Medical University, Chongqing, China, **2** Department of Cardiothoracic Surgery, The First Affiliated Hospital of Chongqing Medical University, Chongqing, China, **3** Thoraxcenter, Department of Cardiology, Erasmus MC University Medical Center, Rotterdam, the Netherlands, **4** Department of Neurosurgery, Banan Hospital Affiliated to Chongqing Medical University, Chongqing, China, **5** Department of Hepatobiliary Surgery, The First Affiliated Hospital of Chongqing Medical University, Chongqing, China, **6** Department of Hepatobiliary Surgery, Bishan Hospital of Chongqing Medical University, Chongqing, China

☯ These authors contributed equally to this work.
* jiangyinjiu@aliyun.com (YJ); taorui1@126.com (RT); wzjtcy@126.com (ZW)

## Abstract

### Background

Alcohol use is a major risk factor for pancreatitis and global mortality. Despite its impact, comprehensive analyses of the burden across regions and sociodemographic strata remain scarce. This study examines global trends (1990–2021) and projects future burdens to 2040.

### Method

Data on Deaths and Disability-Adjusted Life Years (DALYs) attributable to alcohol-induced pancreatitis were retrieved from the 2021 iteration of the Global Burden of Diseases (GBD) database. Trends were analyzed, Age-Period-Cohort models quantified age, period, and cohort effects on Age-Standardized Mortality Rate (ASMR) and Age-Standardized Disability-Adjusted Life Year (ASDR). Associations with SDI were evaluated, burden decomposition applied, and projections made to 2040.

### Results

Globally, both age-standardized death rate (ASDR) and mortality ratio (ASMR) exhibited overall declines (EAPC for ASDR: −0.32%; ASMR: −0.27%), but the reduction was significantly smaller in males (ASDR-EAPC: −0.18%) compared to females (−1.34%). Notablly,Eastern Europe had the highest burden (ASDR: 64.03), High-income Asia-Pacific saw the largest declines (EAPC for ASDR: −2.45%; EAPC

**Data availability statement:** All relevant data are within the manuscript and its Supporting Information files.

**Funding:** The author(s) received no specific funding for this work.

**Competing interests:** The authors have declared that no competing interests exist.

**Abbreviations:** APC, Age-Period-Cohort; ASDR, Age-Standardized Disability-Adjusted Life Year; ASMR, Age-Standardized Mortality Rate; ASR, age-standardized rate; BAPC, Bayesian Age-Period-Cohort; DALYs, Disability-Adjusted Life Years; EAPC, Estimated Annual Percentage Change; CI, confidence interval; GBD, Global Burden of Disease; SDI, Sociodemographic Index; YLD, Years Lived with Disability; YLL, Years of Life Lost.

for ASMR: −2.96%), while Southeast Asia experienced the fastest increase (EAPC for ASDR: 1.98%; EAPC for ASMR: 2.13%). ASDR peaked at ages 45–49, with high-middle SDI countries showing the highest values. Period and cohort effects varied by SDI, with downward trends in high SDI countries and upward trends in lower SDI groups. Population growth and aging drove increases in deaths and DALYs, while epidemiologic changes reduced them. By 2040, deaths and DALYs will stabilize, ASMR will decline until 2034 then rise, and ASDR will decrease until 2035 then increase, driven by population growth.

## Conclusion

Alcohol-related pancreatitis burden demonstrates striking gender, age, and geographic heterogeneity. Targeted policies for high-risk groups (middle-aged/elderly males) and regions (Eastern Europe, Low-middle SDI countries), coupled with preparedness for aging-related burden escalation, are urgently needed.

## 1 Introduction

Pancreatitis is an inflammatory disease of the pancreas, characterized by the abnormal activation of enzymes within the organ, resulting in self-destructive tissue damage. This process triggers an inflammatory response that results in pancreatic dysfunction and may interfere with the normal functioning of distal organs and systems, posing a significant global public health challenge [1]. A comprehensive meta-analysis conducted by Xiao et al. demonstrated that the global annual incidence of acute pancreatitis (AP) was 33.74 per 100,000 individuals (95% CI 23.33–48.81), significantly higher than the incidence of chronic pancreatitis (CP), which was 9.62 per 100,000 (95% CI 7.86–11.78). Additionally, the mortality rate of AP (1.60 per 100,000) was considerably higher than that of CP (0.09 per 100,000), highlighting the acute form's greater lethality [2]. Although AP occurs in nearly equal proportions among men and women, CP, particularly alcohol-related CP, is more prevalent in men [3]. Alcohol, a key modifiable risk factor [4], is the most common cause of CP and the second most common cause of AP after gallstones [5].

Notably, the epidemiologic profile of alcohol-related pancreatitis is closely linked to socioeconomic inequalities. Patients in low-income regions (e.g., Eastern Europe) experience significantly worse prognoses, with an age-standardized disability-adjusted life-years (ASDR) approximately 17 times higher than that of patients in high-income areas (e.g., Asia-Pacific high-income regions) [6]. This disparity is primarily attributed to differences in alcohol consumption patterns, unequal distribution of social resources, limited access to healthcare, and insufficient public health interventions [7,8]. These findings underscore the urgent need to systematically investigate the interaction between socioeconomic inequality and alcohol consumption, with the GBD database providing a valuable resource for such analysis.Despite existing studies on the morbidity burden of pancreatitis, comprehensive investigations into its mortality, disability-adjusted life years (DALYs) and future trends remain scarce,

often relying on relatively outdated data [9]. Moreover, prior research has predominantly been based on single-center or regional cohorts [10,11], which limits sample representativeness and fails to incorporate multidimensional metrics such as DALYs and sociodemographic indices (SDIs). This shortfall hinders a thorough understanding of the global heterogeneity of the disease. The GBD 2021 study integrates national-level data from 204 countries and territories, utilizing a Bayesian hierarchical model (BHM) to address underreporting and bias. This approach enables cross-regional and SDI-stratified analyses of alcohol-related pancreatitis, providing a more robust and comprehensive perspective [6].

Employing the latest data from the GBD 2021, this study applied age-period-cohort (APC) model to quantify the age effect, period effect, and cohort effect on the epidemiological patterns of alcohol-associated pancreatitis, including death, and DALY rates over the period from 1990 to 2021. Additionally, it employed the Bayesian-APC (BAPC) model to project disease trends from 2022 to 2040. To evaluate the global burden, trends of pancreatitis due to alcohol use, the analysis encompassed 204 countries and territories across 21 regions and five sociodemographic index (SDI) subgroups. The findings of this study will offer valuable insights into the complex interplay between alcohol consumption, sociodemographic development, and pancreatitis burden. This evidence-based foundation is crucial for designing targeted public health interventions and policies aimed at mitigating the disease burden of alcohol-induced pancreatitis and enhancing global public health outcomes.

## 2 Methods

### 2.1 Data source

The primary data source for this study was the Global Burden of Disease 2021 (GBD2021), accessed through the Global Health Data Exchange GBD Outcomes Tool (https://ghdx.healthdata.org/gbd-2021). GBD2021 offers a comprehensive assessment of mortality patterns, stratified by age and gender, encompassing 288 causes across 204 countries and 811 localities from 1990 to 2021, along with comparative risks for 88 risk factors [12]. It also evaluates disability-adjusted life expectancy for 371 diseases and injuries. Our analysis specifically focused on extracting GBD2021 datasets pertaining to alcohol-induced pancreatitis during the period from 1990 to 2021. Data included the number of Death cases, age-standardized mortality rate (ASMR), the number of DALY cases, and ASDR. DALYs, a key metric for measuring the public health impact of a disease, combine two components, years of life lost due to premature death (YLL) and years lived with disability (YLD). All rates are reported per 100,000 population.Additionally, the Sociodemographic Index (SDI), which quantifies the economic and sociodemographic status of a country or region, was used to classify the 204 national and regional entities into five categories according to SDI levels: High SDI (0.80–1.0), High-middle SDI (0.70–0.80), Middle SDI (0.60–0.70), Low-middle SDI (0.45–0.60), and Low SDI (0–0.45) [13]. Details regarding the modeling approaches and methodologies employed in GBD2021 are available in related publications [14,15]. This study adheres to the definition and classification criteria of alcoholic pancreatitis as stipulated by the Global Burden of Disease (GBD) study, which refers to acute or chronic pancreatitis caused by long-term alcohol consumption (≥ 4 standard drinks per day for at least 5 years), corresponding to ICD-10 codes K86.0 and K85.2. In the model construction, we integrated the GBD 2019 attribution framework and, through stratified analysis, isolated the association between alcohol exposure and the risk of pancreatitis from other known confounding factors such as cholelithiasis, smoking, hyperlipidemia, and hypercalcemia.

### 2.2 Descriptive and trend analyses

To comprehensively evaluate the global health impact of pancreatitis due to alcohol consumption during the three-decade period spanning 1990–2021, we conducted descriptive analyses at the global, regional, national, and five SDI quintile levels. The primary indicators analyzed included the number of DALY cases, ASDR, the number of Death cases, and ASMR for pancreatitis due to alcohol use. To assess trends in these rates over time, we calculated the estimated annual percentage change (EAPC). EAPC is calculated through a linear regression model, taking into account the variation of the ratio by year,A linear regression model was constructed as $y = \alpha + \beta x$, in which $y = \ln(ASR)$ and $x = $ calendar year. The EAPC

was then calculated by (exp(β)-1) * 100%, with corresponding 95% confidence interval (CI) obtained through the analytical model [16].An ASR was considered to be in a downward trend if both the EAPC value and the upper limit of its 95% CI were < 0; conversely, an upward trend was identified if both the EAPC value and the lower limit of its 95% CI were > 0. The ASR was classified as stable if the absolute value of the EAPC was close to 0. Additionally, we analyzed variations in the disease burden of pancreatitis due to alcohol use between male and female across different age groups to further understand gender-based differences.

### 2.3 Age-period-cohort analysis

Our investigation additionally employed the age-period-cohort model (APC), applied to different SDI levels, which quantifies the independent contributions of age, period, and birth cohort to the ASMR and ASDR in pancreatitis due to alcohol use [17].Age effects are defined as biological changes in disease risk in a given age group, period effects reflect the impact of environmental or policy interventions across years, and cohort effects indicate cumulative exposure to health risks in the same birth cohort population [18].The APC model was obtained through the National Cancer Institute (https://analysistools.nci.nih.gov/apc), and the APC model could be written as a log-linear Poissonmodel: $\ln E_{rij} = \ln \theta_{i-j} N_{ij} = \mu + \alpha_i + \beta_j + \gamma_k$, where $E_{rij}$ represents the expectation for rate, μ the mean effect, and $\alpha_i$, $\beta_j$ and $\gamma_k$ the age, period, and cohort effects, respectively [19].

### 2.4 Relationship between SDI and ASR

To evaluate the associations between SDI and both ASMR and ASDR, Spearman correlation coefficients (r) were computed. The formula for Spearman's r is: $r = 1 - (6\Sigma d_i^2)/ [n(n^2-1)]$, where $d_i$ represents the difference in ranks of corresponding variables and n is the number of observations [20]. The r value ranges from −1–1, with values closer to the extremes indicating stronger correlations. Positive r values signify positive relationships, whereas negative values reflect inverse relationships. Statistical significance was defined as a p-values less than 0.05 as the critical value.

### 2.5 Decomposition analysis

This study utilized the Das Gupta decomposition analysis method to determine the drivers of time-series changes in the number of Death cases, ASMR, the number of DALY cases, and ASDR for pancreatitis due to alcohol use over the period 1990–2021 [21,22]. This approach attributes time-series changes in the burden of death and disability-adjusted life years (DALYs) to three drivers: epidemiological variability, which refers to age-specific changes in disease-specific morbidity or mortality rates, mirroring the effects of medical technology innovations and interventions stemming from public health policies; population size expansion, which refers to the amplifying effect of growth in the total population size on the absolute value of the burden of disease even if age-specific rates of morbidityor mortality rates remain stable; and aging, which reflects the cumulative effect of a rising proportion of older age groups in the population on the burden of chronic and degenerative diseases, the risk of which typically rises significantly with age.

### 2.6 Projection analysis

In order to better formulate the allocation of health care resources and develop health care policies, we made further projections of the global burden of pancreatitis due to alcohol use through 2040. The projections were made using Bayesian Age-Period-Cohort Analysis (BAPC), which integrates integrated nested Laplacian approximation (INLA) for complete Bayesian inference [23]. By introducing the Prior Distribution and Markov Chain Monte Carlo (MCMC) algorithms, the Identifiability Problem, i.e., the covariance problem between age, period, and cohort effects, is solved in the traditional APC model. The covariance problem between age, period and cohort effects is solved in the traditional APC model. The BAPC and INLA packages within R software (v4.4.0) were utilized to conduct these projections. Our implementation enhanced traditional APC modeling through two integrated components: (1) Gaussian process priors with regularization

for cohort effects, effectively balancing overdispersion mitigation and temporal pattern preservation; (2) The Knuiman-Cholesterol constraint system anchored by WHO surveillance trends, imposing biologically plausible boundaries on period slopes via INLA's linear combination sampling.

## 2.7 Ethics approval statement

The GBD 2021 study is an open database, and all data is anonymous. No additional ethical approval is required for this study.

## 3 Result

### 3.1 Global, regional, and national trends and burdens of pancreatitis due to alcohol use

Globally, between 1990 and 2021, both ASDR and ASMR of pancreatitis due to alcohol use showed a declining trend during this period, with EAPC of −0.32% (95% CI: −0.53 to −0.11) and −0.27% (95% CI: −0.45 to −0.08), respectively (Tables 1 and 2). Among males, the EAPC for ASDR and ASMR were −0.18% (95% CI: −0.38 to 0.02) and −0.11% (95% CI: −0.28 to 0.07), respectively, which were significantly smaller declines compared to females, whose EAPC for ASDR and ASMR were −1.34% (95% CI: −1.66 to −1.01) and −1.41% (95% CI: −1.66 to −1.17), respectively (Tables 3–6). The most substantial health burden of alcohol-induced pancreatitis was observed in middle-aged and elderly men worldwide, with number of DALY cases, number of Death cases, ASDR, and ASMR all substantially higher in males than females. Among males, the highest distribution of DALY cases occurred in those aged 35–39 years, while the highest distribution of death cases occurred in those aged 40–69 years. In contrast, in females, the highest ASDR was observed after age 85 (Fig 1).

Regionally, Eastern Europe recorded the highest number of DALY cases, death cases, ASDR, and ASMR, whereas North Africa and Middle East reported the lowest values. The high-income Asia Pacific experienced the most pronounced annual decline in ASDR (−2.45%; 95% CI: −2.61 to −2.29) and ASMR (−2.96%; 95% CI: −3.14 to −2.78). Conversely, Southeast Asia showed the greatest increase in these rates, with EAPC for ASDR at 1.98% (95% CI: 1.87 to 2.08) and for ASMR at 2.13% (95% CI: 2.01 to 2.25) (Tables 1, 2).

Nationally, the burden of pancreatitis due to alcohol use varied widely. Among combined genders (both), males, and females, the Russian Federation had the highest current ASDR at 69.92 (combined; 95% CI: 51.17 to 91.82), 133.67 (males; 95% CI: 99.29 to 174.38), and 14.13 (females; 95% CI: 7.19 to 22.20), respectively (S1-S3 Tables). In terms of EAPC, the EAPC of ASDR is largest for Myanmar and smallest for Bhutan, and the EAPC of ASMR is largest for Cook Islands and smallest for Bhutan (S4 Table).

Sociodemographically,the High-middle SDI quintile had the highest burden in terms of DALY cases, death cases, ASDR, and ASMR, while the lowest burdens were observed in the Low SDI quintile. Notably, countries in the High SDI quintile experienced the fastest declines in ASDR (−1%; 95% CI: −1.13 to −0.88) and ASMR (−0.92%; 95% CI: −1.06 to −0.78). In contrast, the Low-middle SDI quintile exhibited the greatest increases in these metrics, with EAPC for ASDR at 1.55% (95% CI: 1.33 to 1.77) and for ASMR at 1.61% (95% CI: 1.38 to 1.83) (Tables 1, 2).

### 3.2 Age-period-cohort analyses of ASDR and ASMR in pancreatitis due to alcohol use

Age-period-cohort analyses of ASDR and ASMR for pancreatitis due to alcohol use indicated three core findings: Age effects revealed ASDR peaked in the 45–49 age group; Period effects showed continuous declines in High SDI countries but rising trends in Low/Middle SDI groups post-1997. Birth cohort effects highlighted generational ASDR reductions in High SDI nations versus upward trends in Low SDI regions. (Fig 3, Fig 4 and S5 Table). Among these, high-middle SDI showed the highest ASDR, peaking at around 50 in the 45–49 age group (S5 Table), whereas low-middle SDI, as well as low SDI, had the lowest ASDR (Fig 2A). In terms of period effects, the global ASDR demonstrated a pattern of gradual

**Table 1. The case number and ASR of DALYs of pancreatitis due to alcohol use in 1990 and 2021 for both sexes by SDI quintiles and by GBD regions, with EAPC from 1990 to 2021.**

| | 1990 | | 2021 | | EAPC (95%CI) 1990–2021 |
|---|---|---|---|---|---|
| | Number (95%UIs) | ASR (95%UIs) | Number (95%UIs) | ASR (95%UIs) | |
| Global | 401671.32 (280352.07-543581.55) | 8.88 (6.25-12.04) | 699335.04 (486293.01-924031.99) | 8.22 (5.72-10.86) | −0.32 (−0.53 to −0.11) |
| SDI quintiles | | | | | |
| High SDI | 105279.22 (76181.43-137281.6) | 10.29 (7.45-13.42) | 123550.06 (85940.38-165401.01) | 7.81 (5.51-10.27) | −1 (−1.13 to −0.88) |
| High-middle SDI | 168384.71 (121086.91-223703.72) | 15.87 (11.43-21.08) | 251721.93 (179902.73-335940.24) | 14.67 (10.44-19.43) | −0.48 (−0.97 to 0.01) |
| Middle SDI | 75787.96 (48713.62-108007.14) | 5.41 (3.47-7.78) | 167046.39 (115483.75-220626.33) | 6.04 (4.19-7.97) | 0.49 (0.36 to 0.61) |
| Low-middle SDI | 37546.82 (19099.05-62741.23) | 4.46 (2.33-7.46) | 111718.11 (70375.85-154954.43) | 6.29 (3.94-8.8) | 1.55 (1.33 to 1.77) |
| Low SDI | 13867.01 (6506.13-24872.78) | 4.45 (2.11-7.72) | 44400.24 (25590.11-70884.6) | 5.97 (3.44-9.47) | 1.18 (0.98 to 1.38) |
| GBD regions | | | | | |
| Andean Latin America | 3771.09 (1865.61-5788.44) | 13.5 (6.97-20.68) | 7404.49 (4338.62-10793.83) | 11.32 (6.61-16.46) | −0.43 (−0.62 to −0.25) |
| Australasia | 1736.65 (1223.5-2318.98) | 7.64 (5.43-10.16) | 2436.49 (1669.69-3268.66) | 5.38 (3.78-7.11) | −0.89 (−1.11 to −0.66) |
| Caribbean | 2131.29 (1328.29-2991.58) | 7.1 (4.46-10.01) | 3524.06 (2267.28-5121.86) | 6.82 (4.39-9.9) | −0.02 (−0.15 to 0.12) |
| Central Asia | 8372.98 (5202.05-11751.12) | 15.21 (9.28-21.53) | 13675.75 (8901.74-18939.54) | 13.82 (8.97-19.12) | −0.98 (−1.28 to −0.68) |
| Central Europe | 40761.94 (28795.04-52294.83) | 28.87 (20.52-36.94) | 44452.64 (32155.76-56844.34) | 27.25 (20.11-34.73) | −0.5 (−0.72 to −0.28) |
| Central Latin America | 14114.38 (9955.92-18763.85) | 11.57 (8.22-15.36) | 30601.9 (21039.68-41865.7) | 11.47 (7.87-15.69) | −0.22 (−0.5 to 0.05) |
| Central Sub-Saharan Africa | 1185.19 (413.42-2544.65) | 3.58 (1.21-7.55) | 3969.67 (1812.91-7481.1) | 4.41 (2.04-7.88) | 1.57 (0.71 to 2.44) |
| East Asia | 38692.76 (23581.73-56720.47) | 3.61 (2.17-5.32) | 60082.9 (36857.3-90978.81) | 3.02 (1.87-4.56) | −0.47 (−0.73 to −0.21) |
| Eastern Europe | 92294.97 (65457.79-127057.05) | 35.77 (25.42-49.06) | 172784.73 (124114.94-228415.87) | 64.03 (46.21-84.35) | 1.5 (0.68 to 2.33) |
| Eastern Sub-Saharan Africa | 4247.82 (1765.63-7778.16) | 4.04 (1.74-7.34) | 13784.63 (6639.07-23292.3) | 5.21 (2.6-8.6) | 0.85 (0.7 to 0.99) |
| High-income Asia Pacific | 14414.69 (9355.15-19958.12) | 7.15 (4.64-9.92) | 11139.67 (7002.21-15962.18) | 3.75 (2.41-5.38) | −2.45 (−2.61 to −2.29) |
| High-income North Pacific | 23034.51 (16050.8-31514.45) | 7.22 (5.04-9.89) | 37078.47 (23916.67-51449.37) | 7.46 (4.95-10.29) | 0.19 (0.01 to 0.38) |
| North Africa and Middle East | 1011.96 (565.82-1582.79) | 0.46 (0.25-0.73) | 2003.29 (1172.27-2993.95) | 0.36 (0.21-0.55) | −0.89 (−1.01 to −0.77) |
| Oceania | 113.32 (42.77-228.57) | 2.19 (0.82-4.41) | 182.26 (77.79-346.87) | 1.49 (0.63-2.79) | −1.18 (−1.45 to −0.9) |
| South Asia | 33316.01 (14524.98-60398.74) | 4.03 (1.8-7.28) | 97271.35 (56719.04-140892.03) | 5.42 (3.16-7.89) | 1.55 (1.22 to 1.89) |
| Southeast Asia | 11512.13 (6426.85-18402.05) | 3.19 (1.78-5.1) | 41232.82 (25818.48-63715.2) | 5.49 (3.43-8.48) | 1.98 (1.87 to 2.08) |
| Southern Latin America | 11088.74 (7959.66-14354.44) | 23.48 (16.83-30.47) | 9093.92 (6022.68-12540.33) | 11.29 (7.5-15.61) | −1.87 (−2.16 to −1.58) |

*(Continued)*

| | 1990 | | 2021 | | EAPC (95%CI) 1990–2021 |
|---|---|---|---|---|---|
| | Number (95%UIs) | ASR (95%UIs) | Number (95%UIs) | ASR (95%UIs) | |
| Southern Sub-Saharan Africa | 3223.84 (1989.27-4706.8) | 8.54 (5.21-12.73) | 6248.11 (4160.2-8485.93) | 8.29 (5.51-11.16) | −0.1 (−0.24 to 0.04) |
| Tropical Latin America | 17843.03 (12622.23-23988.51) | 14.17 (9.88-19.28) | 36126.77 (24746.33-49177.91) | 13.82 (9.48-18.81) | 0.09 (−0.24 to 0.43) |
| Western Europe | 63947.08 (46291.13-81054.21) | 12.97 (9.55-16.28) | 57699.82 (40009.8-76620.4) | 8.07 (5.72-10.55) | −1.6 (−1.7 to −1.51) |
| Western Sub-Saharan Africa | 14856.94 (7826.06-25967.61) | 12.84 (6.88-22.08) | 48541.29 (30471.14-71775.44) | 16.16 (10.23-23.61) | 0.71 (0.66 to 0.77) |

**Abbreviations:** ASR, age-standardized rate; DALYs, disability-adjusted life-years;SDI, sociodemographic index; GBD, Global Burden of Diseases; EAPC, estimated annual percentage change; CI, confidence interval.

decline followed by a slight increase and subsequent decrease. High SDI mirrored this trend and had the lowest ASDR among the five SDI quintiles, with a continuous downward trend observed. In contrast, middle, low-middle, and low SDI experienced an increase in ASDR starting in the 1997–2001 period, with low-middle SDI recording the highest ASDR during this period (S6 Table; Fig 2B). Regarding birth cohort effects, the global trend in ASDR generally demonstrated a downward trajectory. This downward trend was mirrored in high SDI and high-middle SDI, whereas an upward trend was observed in the remaining SDI groups (S7 Table; Fig 2C). For ASMR, after controlling for period and birth cohort effects, all five SDI quintile groups showed increasing trends with age. The highest ASMR was recorded in high-middle SDI, while low-middle and low SDI had the lowest values (Fig 3). The period and birth cohort effects for ASMR followed patterns similar to those observed for ASDR.

### 3.3  Correlation analysis between SDI and rates

The analysis reveals a non-linear trend: ASDR initially decreases with increasing SDI, then rises gradually, followed by a sharper increase, peaking at approximately 27 when SDI reaches around 0.74. After this peak, ASDR decreases rapidly. From 1990 to 2021, certain regions—such as Western Sub-Saharan Africa, Tropical Latin America, Central Europe, and Western Europe—exhibited ASDR values higher than expected based on their respective SDI. Conversely, Oceania and North Africa and Middle East showed ASDR values lower than expected for their SDI. A positive correlation between ASDR and SDI was observed across regions ($p = 0.30$, $p < 0.001$) (Fig 4). Similarly, Fig 5 demonstrates the relationship between ASMR and SDI across regions, which closely parallels the relationship observed between SDI and ASDR. ASMR was also found to exceed expectations based on SDI in Western Sub-Saharan Africa, Tropical Latin America, Central Europe, Western Europe, and Central Latin America, while it was lower than expected in Oceania and North Africa and Middle East. A positive correlation between ASMR and SDI was observed across regions ($p = 0.32$, $p < 0.001$) (Fig 5).

### 3.4  Decomposition analysis of DALYs and deaths for pancreatitis due to alcohol use

Between 1990 and 2021, global deaths and DALYs attributable to pancreatitis due to alcohol use increased significantly, with the most substantial rises observed in middle-SDI countries (Fig 6). In Fig 6, the black dots represent the overall contributions of three key influencing factors: epidemiologic change, population growth, and aging. Globally, epidemiologic change, population growth, and aging contributed −12.62%, 77.95%, and 34.66%, respectively, to the increase in deaths (S8 Table). In detail, −12.62% Epidemiologic change refers to negative contribution: This indicates that due to improvements in the epidemiological environment (such as reduced alcohol consumption, increased early diagnosis rates,

**Table 2. The case number and ASR of Deaths of pancreatitis due to alcohol use in 1990 and 2021 for both sexes by SDI quintiles and by GBD regions, with EAPC from 1990 to 2021.**

| | 1990 | | 2021 | | EAPC (95%CI) 1990–2021 |
|---|---|---|---|---|---|
| | Number (95%UIs) | ASR (95%UIs) | Number (95%UIs) | ASR (95%UIs) | |
| Global | 9971.79 (6888.25-13403.82) | 0.24 (0.16-0.32) | 18749.03 (12763.28-24677.82) | 0.22 (0.15-0.29) | −0.27 (−0.45 to −0.08) |
| **SDI quintiles** | | | | | |
| High SDI | 3033.32 (2119.88-3998.38) | 0.29 (0.2-0.38) | 4085.43 (2712.54-5502.75) | 0.22 (0.15-0.29) | −0.92 (−1.06 to −0.78) |
| High-middle SDI | 3952.61 (2787.05-5163.82) | 0.39 (0.27-0.51) | 6506.76 (4581.97-8559.82) | 0.36 (0.26-0.47) | −0.42 (−0.85 to 0) |
| Middle SDI | 1748.85 (1114.8-2506.61) | 0.14 (0.09-0.2) | 4343.04 (2846.11-5783.77) | 0.16 (0.1-0.21) | 0.61 (0.49 to 0.74) |
| Low-middle SDI | 879.34 (444.17-1471.55) | 0.12 (0.06-0.19) | 2725.42 (1680.21-3898.78) | 0.16 (0.1-0.24) | 1.61 (1.38 to 1.83) |
| Low SDI | 336.81 (157.62-585.03) | 0.12 (0.06-0.21) | 1061.72 (607.57-1684.83) | 0.16 (0.1-0.26) | 1.21 (1.01 to 1.41) |
| **GBD regions** | | | | | |
| Andean Latin America | 89.79 (46.86-139.31) | 0.36 (0.18-0.56) | 198.14 (110.23-292.16) | 0.32 (0.17-0.47) | −0.23 (−0.43 to −0.02) |
| Australasia | 57.06 (37.31-78.31) | 0.25 (0.16-0.34) | 93.45 (59.4-131.39) | 0.18 (0.12-0.25) | −0.83 (−1.06 to −0.59) |
| Caribbean | 51.1 (31.88-72.19) | 0.18 (0.11-0.26) | 91.39 (54.35-131.23) | 0.17 (0.1-0.25) | 0.01 (−0.15 to 0.17) |
| Central Asia | 192.17 (115.05-274.35) | 0.37 (0.22-0.53) | 308.27 (201-440.18) | 0.33 (0.21-0.47) | −1.07 (−1.38 to −0.77) |
| Central Europe | 1041.46 (714.35-1346.16) | 0.73 (0.5-0.94) | 1341.6 (945.32-1745.7) | 0.73 (0.53-0.94) | −0.22 (−0.46 to 0.01) |
| Central Latin America | 312.85 (220.5-411.63) | 0.29 (0.2-0.38) | 752.12 (486.7-1053.19) | 0.29 (0.18-0.4) | −0.18 (−0.43 to 0.07) |
| Central Sub-Saharan Africa | 28.17 (9.32-59.92) | 0.1 (0.03-0.21) | 90.48 (42.06-164.92) | 0.12 (0.05-0.22) | 1.45 (0.55 to 2.36) |
| East Asia | 950.46 (542.9-1417.08) | 0.1 (0.06-0.15) | 1823.7 (1072.84-2814.59) | 0.09 (0.05-0.14) | −0.22 (−0.45 to 0.01) |
| Eastern Europe | 1744.85 (1237.39-2265.95) | 0.66 (0.48-0.86) | 3868.69 (2756.17-5101.41) | 1.35 (0.98-1.78) | 1.91 (1.08 to 2.73) |
| Eastern Sub-Saharan Africa | 104.75 (44.38-192.42) | 0.12 (0.05-0.22) | 326 (162.84-543.28) | 0.15 (0.07-0.24) | 0.75 (0.61 to 0.9) |
| High-income Asia Pacific | 362.56 (222.17-492.7) | 0.18 (0.11-0.25) | 352.76 (208.69-501.62) | 0.09 (0.05-0.12) | −2.96 (−3.14 to −2.78) |
| High-income North Pacific | 585.73 (390.68-819.15) | 0.18 (0.12-0.24) | 1116.22 (686.57-1557.12) | 0.2 (0.13-0.27) | 0.47 (0.22 to 0.71) |
| North Africa and Middle East | 23.72 (12.4-38.16) | 0.01 (0.01-0.02) | 51.09 (26.6-80.88) | 0.01 (0.01-0.02) | −0.71 (−0.83 to −0.59) |
| Oceania | 2.24 (0.82-4.6) | 0.05 (0.02-0.1) | 3.71 (1.52-7.13) | 0.03 (0.01-0.06) | −1.11 (−1.36 to −0.85) |
| South Asia | 767.5 (342.61-1394.36) | 0.1 (0.04-0.19) | 2376.16 (1355.6-3514.75) | 0.14 (0.08-0.21) | 1.66 (1.3 to 2.02) |
| Southeast Asia | 263.03 (146.37-425.65) | 0.08 (0.04-0.13) | 1032.36 (629.69-1618.33) | 0.15 (0.09-0.23) | 2.13 (2.01 to 2.25) |
| Southern Latin America | 339.34 (235.08-443.95) | 0.73 (0.51-0.96) | 298.45 (188.62-425.41) | 0.35 (0.23-0.5) | −1.81 (−2.09 to −1.53) |

*(Continued)*

**Table 2.** (Continued)

| | 1990 | | 2021 | | EAPC (95%CI) 1990–2021 |
|---|---|---|---|---|---|
| | Number (95%UIs) | ASR (95%UIs) | Number (95%UIs) | ASR (95%UIs) | |
| Southern Sub-Saharan Africa | 72.52 (42.79-111.32) | 0.22 (0.12-0.34) | 147.49 (97.24-201.15) | 0.21 (0.14-0.29) | −0.01 (−0.12 to 0.09) |
| Tropical Latin America | 401.97 (276.65-547.28) | 0.35 (0.24-0.48) | 957.88 (637.64-1297.54) | 0.37 (0.24-0.5) | 0.42 (0.06 to 0.78) |
| Western Europe | 2209.67 (1565.66-2870.75) | 0.41 (0.29-0.53) | 2343.24 (1540.15-3172.73) | 0.26 (0.18-0.35) | −1.43 (−1.52 to −1.34) |
| Western Sub-Saharan Africa | 370.87 (198.93-632.55) | 0.36 (0.19-0.6) | 1175.84 (739.52-1714.22) | 0.45 (0.28-0.65) | 0.76 (0.71 to 0.82) |

**Abbreviations:** ASR, age-standardized rate; DALYs, disability-adjusted life-years;SDI, sociodemographic index; GBD, Global Burden of Diseases; EAPC, estimated annual percentage change; CI, confidence interval.

improved treatment options, and the implementation of public health intervention measures), the ASMR of pancreatitis has decreased, thereby partially offsetting the increased burden brought about by population growth and aging.

The contributions of these factors varied substantially across SDI quintile groups. High SDI experienced the largest impact of epidemiologic change (−87.41%), middle SDI showed the greatest influence of population growth (61.82%), and high-middle SDI demonstrated the most pronounced effect of aging (55.48%) (S8 Table). Similarly, for global DALYs, epidemiologic change, population growth, and aging accounted for −14.11%, 88.85%, and 25.26% of the total increase, respectively (S8 Table). The largest contribution of epidemiologic change was again observed in high SDI (−169.14%). Population growth had the greatest impact in high-middle SDI(76.55%), while aging contributed most significantly in high-middle SDI as well (45.61%) (S8 Table).

### 3.5  Predictive analysis on the burden of pancreatitis due to alcohol use to 2040

We projected the number of death cases, ASMR, number of DALY cases and ASDR for pancreatitis due to alcohol use to 2040. Globally, the number of death cases is expected to decline from 2021 until 2025, after which it is projected to stabilize at approximately 1,800 cases per year (Fig 7). Similarly, the ASMR is projected to decline from 2021, reaching a minimum in 2034, followed by a gradual increase thereafter (S9 Table). The number of DALY cases is projected to rise gradually from 2021, stabilize temporarily, and then decline slightly (Fig 7). Meanwhile, the ASDR is expected to decrease gradually until 2035, after which it will begin to rise gradually (S9 Table). The observed decline in ASR alongside rising case numbers may be attributed to global population growth.

## 4  Discussion

Alcohol consumption constitutes a significant risk factor for both acute and chronic pancreatitis, with its associated disease burden and mortality demonstrating substantial heterogeneity across global populations. Utilizing data from the GBD 2021 database, this study integrates DALYs and mortality metrics to analyze trends in alcohol-related pancreatitis burden across global and SDI-stratified regions from 1990 to 2021. We further investigate the impact of socioeconomic, policy, and cultural factors on disease distribution and propose targeted prevention strategies. Globally, the burden of pancreatitis due to alcohol use presents a complex and dynamic landscape. Over the past three decades, while high-income countries have achieved reductions in disease burden through public health interventions, worsening trends in low- and middle-income regions remain concerning [24–26]. This phenomenon reflects the interplay of multiple factors, including alcohol consumption patterns, socioeconomic transitions, policy implementation efficacy, and biological mechanisms.

**Table 3. The case number and ASR of DALYs of pancreatitis due to alcohol use in 1990 and 2021 for male by SDI quintiles and by GBD regions, with EAPC from 1990 to 2021.**

| | 1990 | | 2021 | | EAPC (95%CI) 1990–2021 |
|---|---|---|---|---|---|
| | Number (95%UIs) | ASR (95%UIs) | Number (95%UIs) | ASR (95%UIs) | |
| Global | 351071.77 (248861.16-466482.76) | 15.7 (11.2-20.86) | 631610.83 (449579.68-821562.69) | 15.11 (10.76-19.66) | −0.18 (−0.38 to 0.02) |
| SDI quintiles | | | | | |
| High SDI | 84920.31 (62792.46-108571.13) | 17.56 (12.99-22.45) | 100750.97 (71452.75-130659.45) | 13.12 (9.53-16.94) | −1.07 (−1.2 to −0.93) |
| High-middle SDI | 144356.95 (106760.37-189693.10) | 28.27 (21.01-37) | 223496.84 (163766.52-293344.49) | 26.48 (19.43-34.79) | −0.43 (−0.9 to 0.04) |
| Middle SDI | 71800.42 (46660.88-101445.19) | 10.15 (6.61-14.4) | 157770.64 (110885.14-207973.67) | 11.56 (8.13-15.22) | 0.57 (0.44 to 0.7) |
| Low-middle SDI | 36178.54 (18359.59-60720.12) | 8.42 (4.37-14.13) | 106852.22 (66535.11-147931.6) | 12.16 (7.48-16.91) | 1.64 (1.42 to 1.87) |
| Low SDI | 13097.40 (6051.61-24099.71) | 8.27 (3.92-14.87) | 41923.07 (23804.74-67815.37) | 11.33 (6.53-18.26) | 1.24 (1.03 to 1.46) |
| GBD regions | | | | | |
| Andean Latin America | 3378.33 (1803.94-5058.92) | 24.78 (13.69-37.04) | 6666.01 (4077.41-9378.19) | 20.86 (12.6-29.36) | −0.45 (−0.62 to −0.28) |
| Australasia | 1326.69 (966.14-1725.06) | 12.32 (8.93-16.05) | 1886.11 (1323.75-2457.08) | 8.76 (6.24-11.32) | −0.88 (−1.13 to −0.62) |
| Caribbean | 1946.15 (1253.16-2710.06) | 13.38 (8.66-18.64) | 3225.22 (2121.08-4594.51) | 12.83 (8.48-18.35) | −0.04 (−0.18 to 0.1) |
| Central Asia | 7414.41 (4740.12-10089.03) | 28.66 (18.44-39.53) | 12698.69 (8530.83-17287.37) | 26.8 (17.85-36.61) | −0.83 (−1.12 to −0.54) |
| Central Europe | 36485.6 (26463.76-45737.14) | 53.99 (39.25-67.68) | 40446.33 (29953.98-50956.18) | 51.54 (38.64-64.79) | −0.48 (−0.72 to −0.25) |
| Central Latin America | 13407.45 (9786.25-17567.43) | 22.67 (16.6-29.68) | 28707.66 (20283.45-38844.51) | 22.61 (15.93-30.61) | −0.21 (−0.48 to 0.06) |
| Central Sub-Saharan Africa | 1147.97 (407.68-2460.64) | 7.28 (2.43-15.31) | 3841 (1762.99-7237.17) | 8.91 (4.14-15.97) | 1.52 (0.66 to 2.39) |
| East Asia | 36719.23 (22579.66-53764.08) | 6.8 (4.1-9.97) | 57536.77 (35718.79-85710.41) | 5.87 (3.62-8.81) | −0.35 (−0.61 to −0.1) |
| Eastern Europe | 79272.12 (58981.51-106005.76) | 67.45 (50.06-90.37) | 153445.91 (112625.19-200738.47) | 121.94 (90.29-158.56) | 1.57 (0.77 to 2.37) |
| Eastern Sub-Saharan Africa | 4099.92 (1713.44-7497.48) | 7.91 (3.4-14.48) | 13311.13 (6444.88-22515.3) | 10.37 (5.15-17.15) | 0.9 (0.75 to 1.04) |
| High-income Asia Pacific | 11994.37 (7932.13-16389.74) | 12.46 (8.18-17.06) | 9346.07 (6056.94-13328.09) | 6.48 (4.25-9.25) | −2.47 (−2.63 to −2.32) |
| High-income North Pacific | 17757.15 (12468.94-23682.06) | 11.78 (8.22-15.68) | 27963.87 (18626.29-37500.36) | 11.75 (8.13-15.73) | 0.07 (−0.14 to 0.27) |
| North Africa and Middle East | 923.55 (519.45-1425.09) | 0.82 (0.45-1.27) | 1893.47 (1116.04-2829.78) | 0.66 (0.38-1) | −0.78 (−0.91 to −0.66) |
| Oceania | 111.18 (42.13-225.05) | 4.16 (1.54-8.39) | 179.09 (77.3-337.27) | 2.86 (1.22-5.38) | −1.15 (−1.43 to −0.88) |
| South Asia | 32682.21 (14310.7-59694.99) | 7.54 (3.38-13.69) | 94799.79 (54707.23-137667.67) | 10.52 (6.05-15.32) | 1.69 (1.35 to 2.02) |
| Southeast Asia | 11202.47 (6255.81-18019.82) | 6.38 (3.55-10.27) | 40343.03 (25361.54-61692.43) | 10.99 (6.89-16.75) | 1.99 (1.88 to 2.1) |
| Southern Latin America | 8381.12 (6141.59-10655.44) | 37.95 (27.7-48.26) | 7104.34 (4920.31-9685.28) | 18.76 (12.99-25.52) | −1.79 (−2.06 to −1.52) |

*(Continued)*

| | 1990 | | 2021 | | EAPC (95%CI) 1990–2021 |
|---|---|---|---|---|---|
| | Number (95%UIs) | ASR (95%UIs) | Number (95%UIs) | ASR (95%UIs) | |
| Southern Sub-Saharan Africa | 3009.55 (1865.06-4432.35) | 16.95 (10.22-25.92) | 5854.03 (3904.4-7988.82) | 16.55 (11.01-22.34) | −0.1 (−0.26 to 0.05) |
| Tropical Latin America | 16393.07 (11758.33-21885.16) | 26.79 (19.05-35.86) | 31938.65 (22608.21-42484.73) | 25.62 (18.08-34.07) | 0.04 (−0.3 to 0.38) |
| Western Europe | 49620.24 (37428.88-61614.13) | 21.99 (16.65-27.26) | 45685.71 (32459.91-58746.08) | 13.64 (9.91-17.4) | −1.59 (−1.69 to −1.5) |
| Western Sub-Saharan Africa | 13799 (7211.64-24847.32) | 22.44 (11.74-39.78) | 44737.96 (28352.48-66337.48) | 30.62 (19.6-44.57) | 1 (0.94 to 1.06) |

**Abbreviations:** ASR, age-standardized rate; DALYs, disability-adjusted life-years;SDI, sociodemographic index; GBD, Global Burden of Diseases; EAPC, estimated annual percentage change; CI, confidence interval.

Compared with previous study published by Xiao et al. [2], our study advances the field through a dedicated focus on alcohol as a modifiable and preventable risk factor, which previous studies have not isolated. By leveraging GBD 2021 data, we eliminated confounding from other etiological contributors (e.g., biliary, metabolic) to specifically quantify alcohol's role—a critical distinction given alcohol's unique preventability through policy interventions. In addition, our decomposition analysis further revealed how population growth, aging, and epidemiological changes interact to shape burden trends, providing granular insights absent in aggregated analyses. Additionally, our integration of 30-year historical trends (1990–2021) with BAPC-projected trajectories to 2040 offers actionable foresight for policymakers, particularly in emerging hotspots like South Asia. These alcohol-specific, forward-looking analyses uniquely inform targeted resource allocation—a key advancement beyond descriptive burden reporting.

From 1990 to 2021, global DALYs attributed to alcohol-related pancreatitis increased from 401,000–699,000 (74% rise). Decomposition analysis suggests this growth was primarily driven by population expansion and aging—global population increased by 46% during this period, with the proportion of individuals aged ≥65 years rising from 6% to 9.3% [13]. However, the decline in ASR (from 8.88 to 8.22 per 100,000; estimated annual percentage change [EAPC] = −0.32%) indicates partial mitigation of demographic impacts through public health measures. This paradoxical pattern of "An increase in absolute numbers juxtaposed with a decline in relative rates" highlights coexisting achievements and persistent challenges in disease control. Notably, gender disparity remains a defining characteristic of disease burden. In 2021, males accounted for 631,600 DALYs—9.3 times higher than females (67,724 DALYs)—a gap strongly associated with higher alcohol consumption (global averages: 19.6 liters for males vs. 7.0 liters for females) and prevalence of high-risk drinking behaviors (e.g., heavy episodic drinking). Males aged 50–69 years bore 62% of the total burden, underscoring the cumulative effects of prolonged alcohol exposure. Conversely, females surpassed males in burden among those aged ≥80 years, potentially linked to postmenopausal estrogen depletion and increased comorbidities (e.g., diabetes) [27–29]. Projections using the BAPC model suggest persistent gender disparities in disease burden by 2040. For males, ASDR is projected to decline from 15.11 (2021) to 12.29 per 100,000 (EAPC = −0.52%), while female ASR will decrease from 1.53 to 1.07 per 100,000 (EAPC = −1.18%).

Our findings reveal a steeper decline in alcohol-related pancreatitis burden among females compared to males, despite the persistent overall male predominance. This disparity may reflect multifactorial drivers: 1. Behavioral Factors: Females generally exhibit lower alcohol consumption levels and reduced engagement in high-risk drinking patterns [30], limiting cumulative exposure to alcohol's pancreatotoxic effects; 2. Biological Buffering: Estrogen's anti-inflammatory properties may attenuate pancreatitis severity in premenopausal women, though postmenopausal declines in estrogen correlate with rising burden among elderly females [31]; 3.Policy Targeting: Recent alcohol-control campaigns (e.g., pregnancy-related

**Table 4. The case number and ASR of DALYs of pancreatitis due to alcohol use in 1990 and 2021 for female by SDI quintiles and by GBD regions, with EAPC from 1990 to 2021.**

| | 1990 | | 2021 | | EAPC (95%CI) 1 990–2021 |
|---|---|---|---|---|---|
| | Number (95%UIs) | ASR (95%UIs) | Number (95%UIs) | ASR(95%UIs) | |
| Global | 50599.55 (28543.87-75963.02) | 2.28 (1.28-3.4) | 67724.21 (38051.37-102581.55) | 1.53 (0.86-2.32) | −1.34 (−1.66 to −1.01) |
| SDI quintiles | | | | | |
| High SDI | 20358.91 (12278.48-29736.01) | 3.6 (2.18-5.28) | 22799.1 (13003.28-34373.4) | 2.58 (1.51-3.83) | −1.13 (−1.2 to −1.06) |
| High-middle SDI | 24027.77 (13739.91-36534.6) | 4.35 (2.49-6.62) | 28225.1 (14856.56-43026.78) | 3.12 (1.68-4.75) | −1.2 (−1.81 to −0.59) |
| Middle SDI | 3987.54 (1721.31-6700.58) | 0.6 (0.26-1.01) | 9275.75 (4787.41-14533.17) | 0.67 (0.35-1.05) | 0.42 (0.29 to 0.55) |
| Low-middle SDI | 1368.27 (476.81-2632.87) | 0.36 (0.13-0.69) | 4865.89 (2473.91-8145.47) | 0.57 (0.28-0.96) | 1.86 (1.68 to 2.05) |
| Low SDI | 769.61 (296.17-1526.3) | 0.57 (0.22-1.11) | 2477.17 (1183.27-4201.98) | 0.76 (0.36-1.28) | 1.15 (1.06 to 1.24) |
| GBD regions | | | | | |
| Andean Latin America | 392.76 (3.85-911.56) | 2.63 (0.12-5.86) | 738.49 (196.49-1434.1) | 2.22 (0.55-4.3) | −0.19 (−0.59 to 0.21) |
| Australasia | 409.96 (252.28-602.01) | 3.36 (2.07-4.95) | 550.38 (308.06-859.71) | 2.22 (1.22-3.41) | −1.05 (−1.21 to −0.89) |
| Caribbean | 185.14 (70.81-321.35) | 1.14 (0.42-1.99) | 298.84 (117.03-490.77) | 1.14 (0.46-1.87) | 0.18 (0.08 to 0.28) |
| Central Asia | 958.57 (366.12-1646.37) | 3.32 (1.24-5.67) | 977.06 (385.51-1723.11) | 1.94 (0.78-3.44) | −2.8 (−3.2 to −2.4) |
| Central Europe | 4276.34 (1972.73-6963.78) | 5.52 (2.57-8.96) | 4006.31 (1941.67-6391.96) | 4.08 (1.98-6.49) | −1.08 (−1.27 to −0.89) |
| Central Latin America | 706.93 (204.54-1308.58) | 1.07 (0.32-1.98) | 1894.25 (735.2-3338.41) | 1.37 (0.53-2.41) | 0.5 (0.24 to 0.75) |
| Central Sub-Saharan Africa | 37.22 (5.47-91.06) | 0.23 (0.04-0.55) | 128.67 (42.13-258.54) | 0.3 (0.1-0.6) | 1.91 (1.25 to 2.57) |
| East Asia | 1973.53 (659.02-3833.88) | 0.4 (0.14-0.78) | 2546.13 (1074.71-4503.65) | 0.24 (0.1-0.42) | −1.62 (−2.02 to −1.22) |
| Eastern Europe | 13022.85 (6953.94-21235.57) | 8.96 (4.7-14.54) | 19338.82 (9989.84-30088.03) | 12.91 (6.87-20.04) | 0.97 (0 to 1.95) |
| Eastern Sub-Saharan Africa | 147.9 (55.03-291.63) | 0.29 (0.12-0.57) | 473.51 (210.89-876.54) | 0.38 (0.17-0.71) | 0.96 (0.85 to 1.08) |
| High-income Asia Pacific | 2420.33 (1323.74-3563.6) | 2.3 (1.28-3.38) | 1793.6 (922.21-2757.07) | 1.06 (0.6-1.64) | −2.9 (−3.1 to −2.71) |
| High-income North Pacific | 5277.36 (3180.63-8117.23) | 3.1 (1.88-4.79) | 9114.59 (4935.9-13721.8) | 3.4 (1.95-5.06) | 0.41 (0.29 to 0.53) |
| North Africa and Middle East | 88.4 (33.48-170.82) | 0.09 (0.03-0.17) | 109.82 (42.8-197.59) | 0.05 (0.02-0.08) | −2.69 (−2.96 to −2.42) |
| Oceania | 2.14 (0.39-5.03) | 0.08 (0.02-0.2) | 3.18 (0.42-7.19) | 0.06 (0.01-0.13) | −1.28 (−1.73 to −0.82) |
| South Asia | 633.81 (171.55-1516.69) | 0.17 (0.05-0.41) | 2471.56 (1104.16-4517.29) | 0.29 (0.12-0.53) | 2.47 (2.09 to 2.86) |
| Southeast Asia | 309.66 (134.66-566.35) | 0.19 (0.08-0.36) | 889.79 (471.63-1506.1) | 0.24 (0.13-0.41) | 0.54 (0.39 to 0.68) |
| Southern Latin America | 2707.62 (1726.99-3890.49) | 10.75 (6.86-15.43) | 1989.59 (1029.6-3077.37) | 4.66 (2.48-7.22) | −2.18 (−2.58 to −1.78) |

*(Continued)*

**Table 4.** (Continued)

| | 1990 | | 2021 | | EAPC (95%CI) 1 990–2021 |
|---|---|---|---|---|---|
| | Number (95%UIs) | ASR (95%UIs) | Number (95%UIs) | ASR(95%UIs) | |
| Southern Sub-Saharan Africa | 214.29 (120.43-333.26) | 1.15 (0.63-1.77) | 394.08 (213.9-616.79) | 1.06 (0.56-1.66) | −0.15 (−0.3 to 0) |
| Tropical Latin America | 1449.96 (767.81-2308.7) | 2.28 (1.2-3.67) | 4188.12 (2198.2-6593.46) | 3.11 (1.65-4.88) | 1.06 (0.72 to 1.4) |
| Western Europe | 14326.84 (9048.93-19726.56) | 4.92 (3.17-6.86) | 12014.11 (6834.99-17861.16) | 2.85 (1.71-4.18) | −1.92 (−2.02 to −1.82) |
| Western Sub-Saharan Africa | 1057.94 (411.22-2075.21) | 2.23 (0.87-4.36) | 3803.34 (1745.88-6342.33) | 3.07 (1.43-5.18) | 1.04 (0.94 to 1.14) |

**Abbreviations:** ASR, age-standardized rate; DALYs, disability-adjusted life-years;SDI, sociodemographic index; GBD, Global Burden of Diseases; EAPC, estimated annual percentage change; CI, confidence interval.

drinking warnings) disproportionately reach female populations, amplifying their preventive impact [32]. These sex-specific dynamics highlight the need for gender-tailored interventions—for example, leveraging females' healthcare engagement to promote household-level alcohol reduction, while addressing male-dominated high-risk drinking cultures through workplace-based programs.

The burden of alcohol-induced pancreatitis also exhibits significant regional disparities that cannot be overlooked. Eastern Europe bears the heaviest global burden, with ASDR surging from 35.77 to 64.03 per 100,000 (Estimated Annual Percentage Change [EAPC]=1.5%) and mortality rates (ASMR) escalating from 0.66 to 1.35 per 100,000 (EAPC = 1.91%) – the fastest growth rate globally. Russia and Moldova, in particular, are the countries with the highest burden in Eastern Europe and even the world.This trend is likely driven by high-alcohol-consumption cultures (e.g., widespread spirits consumption) and delayed public health interventions [8]. In contrast, high-income Asia-Pacific regions have achieved remarkable success through comprehensive strategies including increased alcohol taxes, nighttime sales restrictions, and public health education. These measures reduced DALY rates from 7.15 to 3.75 per 100,000 (EAPC=−2.45%) and mortality rates from 0.18 to 0.09 per 100,000 (EAPC=−2.96%), establishing a global model for effective control [33]. North Africa and the Middle East maintain relatively low burdens, with DALY rates declining from 0.46 to 0.36 per 100,000 (EAPC=−0.89%) and stable mortality rates at 0.01 per 100,000 (EAPC=−0.71%), attributable to religious alcohol restrictions and demographic factors [34–36]. However, emerging concerns focus on potential alcohol consumption growth and its long-term impacts [37]. Emerging hotspots appear in South and Southeast Asia, where DALY cases surged by 192% and 258% respectively, with ASDR increasing 34% and 72% (EAPC = 1.55% and 1.98%). Mortality ASR rose 88% and 66%, exacerbated by alcohol market globalization (beer and premixed drinks proliferation) and inadequate control systems amidst rapid economic development [38].Sub-Saharan Africa, particularly West and Central Africa, shows significant ASR increases in both DALYs and mortality, reflecting alcohol accessibility, inadequate health education campaigns, and malnutrition-related public health challenges [39–41].

From the perspective of alcohol consumption policies in various countries, high – income regions such as Western Europe and Australia have significantly reduced the disease burden through comprehensive alcohol control policies. For instance the "Évin Act" promulgated in France in 1989 strictly restricted alcohol advertising, leading to a decline in alcohol – related hospitalization rates [42]. In the UK, the "Minimum Unit Pricing" (MUP) policy implemented in 2018 decreased the sales of cheap spirits by 20%, and also led to a decline in the incidence of alcohol – related diseases and an improvement in prognosis [43]. These policies not only rely on legislation but also change society's perception of alcohol through cultural reshaping, such as the "sober curious" movement in Australia [44].However, upper – middle – income countries still need to address the increase in the absolute number of cases brought about by aging. Take Japan, for example,

**Table 5. The case number and ASR of Deaths of pancreatitis due to alcohol use in 1990 and 2021 for male by SDI quintiles and by GBD regions, with EAPC from 1990 to 2021.**

| | 1990 | | 2021 | | EAPC (95%CI) 1990–2021 |
|---|---|---|---|---|---|
| | Number (95%UIs) | ASR (95%UIs) | Number (95%UIs) | ASR (95%UIs) | |
| Global | 8444.09 (5964.01-11224.16) | 0.42 (0.3-0.56) | 16563.45 (11588-21731) | 0.41 (0.29-0.53) | −0.11 (−0.28 to 0.07) |
| SDI quintiles | | | | | |
| High SDI | 2318.84 (1679.7-2981.37) | 0.5 (0.36-0.64) | 3190.55 (2199.17-4199.8) | 0.37 (0.26-0.49) | −0.98 (−1.13 to −0.84) |
| High-middle SDI | 3302.6 (2423.27-4229.2) | 0.71 (0.53-0.92) | 5690.02 (4160.42-7432.16) | 0.67 (0.49-0.87) | −0.37 (−0.77 to 0.04) |
| Middle SDI | 1647.24 (1068.67-2361.24) | 0.27 (0.17-0.38) | 4077.83 (2724.05-5427.79) | 0.31 (0.21-0.42) | 0.71 (0.58 to 0.84) |
| Low-middle SDI | 842.67 (423.46-1419.45) | 0.22 (0.11-0.36) | 2587.9 (1577.92-3680.37) | 0.32 (0.19-0.46) | 1.74 (1.5 to 1.97) |
| Low SDI | 314.69 (146.72-563.72) | 0.22 (0.1-0.4) | 993.53 (566.12-1602.63) | 0.31 (0.18-0.48) | 1.31 (1.08 to 1.53) |
| GBD regions | | | | | |
| Andean Latin America | 80.85 (44.37-121.89) | 0.66 (0.37-1.02) | 178.029 (103.08-256.57) | 0.59 (0.33-0.85) | −0.23 (−0.42 to −0.05) |
| Australasia | 41.77 (28.2-55.07) | 0.41 (0.28-0.54) | 69.07 (45.99-92.71) | 0.29 (0.19-0.38) | −0.92 (−1.19 to −0.65) |
| Caribbean | 47.01 (29.85-66.07) | 0.34 (0.22-0.49) | 84.06 (51.96-120.89) | 0.33 (0.21-0.48) | 0.01 (−0.15 to 0.18) |
| Central Asia | 167.04 (105.88-231.48) | 0.71 (0.44-0.99) | 284.43 (189.12-399.87) | 0.65 (0.43-0.92) | −0.9 (−1.18 to −0.61) |
| Central Europe | 908.41 (650.86-1146.17) | 1.38 (0.99-1.74) | 1189.41 (867-1524.76) | 1.42 (1.05-1.8) | −0.17 (−0.42 to 0.08) |
| Central Latin America | 297.75 (216.31-387.55) | 0.56 (0.39-0.74) | 704.52 (467.84-959.34) | 0.57 (0.38-0.78) | −0.14 (−0.38 to 0.11) |
| Central Sub-Saharan Africa | 27.43 (9.01-58.22) | 0.21 (0.07-0.44) | 87.91 (40.88-160.38) | 0.25 (0.11-0.46) | 1.52 (0.64 to 2.41) |
| East Asia | 889.47 (517.9-1322.04) | 0.2 (0.11-0.3) | 1729 (1013.08-2672.72) | 0.18 (0.11-0.28) | −0.1 (−0.33 to 0.12) |
| Eastern Europe | 1492.89 (1090.84-1885.26) | 1.31 (0.97-1.67) | 3441.37 (2511.26-4481.95) | 2.67 (1.97-3.49) | 1.99 (1.2 to 2.78) |
| Eastern Sub-Saharan Africa | 101.43 (43.3-187.66) | 0.23 (0.1-0.43) | 315.55 (156.64-524.34) | 0.29 (0.15-0.48) | 0.84 (0.69 to 0.99) |
| High-income Asia Pacific | 290.59 (182.79-387.13) | 0.33 (0.2-0.44) | 273.14 (168.13-379.89) | 0.15 (0.1-0.21) | −2.93 (−3.09 to −2.76) |
| High-income North Pacific | 447.61 (301.41-613.01) | 0.3 (0.2-0.41) | 823.64 (534.76-1113.99) | 0.31 (0.21-0.42) | 0.24 (−0.01 to 0.49) |
| North Africa and Middle East | 21.37 (11.36-33.61) | 0.02 (0.01-0.04) | 47.51 (25.64-75.61) | 0.02 (0.01-0.03) | −0.65 (−0.78 to −0.52) |
| Oceania | 2.2 (0.8-4.52) | 0.09 (0.03-0.19) | 3.65 (1.51-7) | 0.06 (0.03-0.12) | −1.09 (−1.35 to −0.83) |
| South Asia | 751.43 (336.85-1375.32) | 0.19 (0.08-0.35) | 2308.56 (1293.3-3397.18) | 0.28 (0.15-0.41) | 1.83 (1.47 to 2.19) |
| Southeast Asia | 254.54 (141.44-412.99) | 0.16 (0.09-0.27) | 1006.4 (616.19-1557.9) | 0.3 (0.18-0.47) | 2.22 (2.09 to 2.35) |
| Southern Latin America | 255.76 (181.98-328.54) | 1.21 (0.86-1.56) | 230.43 (155.08-318.73) | 0.61 (0.41-0.84) | −1.72 (−1.98 to −1.47) |

*(Continued)*

**Table 5.** (Continued)

| | 1990 | | 2021 | | EAPC (95%CI) 1990–2021 |
|---|---|---|---|---|---|
| | Number (95%UIs) | ASR (95%UIs) | Number (95%UIs) | ASR (95%UIs) | |
| Southern Sub-Saharan Africa | 67.46 (40.07-104.72) | 0.44 (0.25-0.71) | 136.9 (89.62-185.22) | 0.44 (0.29-0.59) | 0 (−0.12 to 0.11) |
| Tropical Latin America | 367.65 (258.42-497.32) | 0.66 (0.46-0.89) | 842.06 (580.59-1129.49) | 0.7 (0.48-0.94) | 0.42 (0.05 to 0.79) |
| Western Europe | 1594.92 (1186.03-2017.76) | 0.7 (0.53-0.89) | 1749.23 (1202.25-2310.09) | 0.45 (0.31-0.59) | −1.44 (−1.54 to −1.34) |
| Western Sub-Saharan Africa | 336.5 (176.95-597.24) | 0.62 (0.33-1.1) | 1058.58 (672.43-1535.43) | 0.84 (0.52-1.21) | 0.98 (0.93 to 1.03) |

**Abbreviations:** ASR, age-standardized rate; DALYs, disability-adjusted life-years;SDI, sociodemographic index; GBD, Global Burden of Diseases; EAPC, estimated annual percentage change; CI, confidence interval.

With a relatively high proportion of the aging population, while the ASR of pancreatitis deaths has decreased, the actual number of pancreatitis deaths has still increased [45,46]. The paradoxical increase in alcohol-related pancreatitis burden within certain mid-SDI regions, despite measurable socioeconomic development, may reflect transitional epidemiological dynamics. As these regions industrialize, accelerated urbanization and disposable income growth often coincide with rising alcohol accessibility, aggressive alcohol marketing, and delayed regulatory policies targeting harmful consumption patterns. Concurrently, dietary shifts toward high-fat, processed foods compound alcohol's metabolic toxicity, exacerbating pancreatic injury. Critically, healthcare systems in mid-SDI settings may prioritize infectious disease control over chronic condition management, leading to underdiagnosis of early-stage pancreatitis and inadequate preventive interventions.

The pathogenesis of alcoholic pancreatitis involves multilevel metabolic disorders and tissue damage mechanisms. At the molecular level, acetaldehyde, which is generated from the metabolism of ethanol by alcohol dehydrogenase in pancreatic acinar cells, not only promotes the premature activation of trypsinogen by activating lysosomal enzymes but also leads to the abnormal accumulation of zymogen granules by inhibiting the autophagy – lysosome system [47,48]. This dual effect disrupts the spatio – temporal order of physiological zymogen activation in the pancreas, triggering the autodigestive process of the pancreatic enzyme cascade activation.Meanwhile, the covalent binding of acetaldehyde to mitochondrial membrane proteins interferes with the function of the electron transport chain, resulting in increased production of reactive oxygen species (ROS) [49]. Moreover, the alcohol metabolism process continuously consumes antioxidant substances such as glutathione, causing the level of oxidative stress to exceed the cellular defense threshold [50]. This imbalance between oxidation and antioxidation not only directly damages mitochondrial DNA but also activates pro – inflammatory signaling pathways such as NF-κB, promoting the transformation of pancreatic stellate cells into myofibroblasts and initiating the fibrosis process [51]. Animal experiments have shown that when animals are in a state of excessive alcohol consumption, pancreatic acinar cells are chronically exposed to the above – mentioned pathological environment. Their mitochondrial functional reserve is gradually exhausted, and endoplasmic reticulum stress is continuously activated, ultimately leading to a mixed injury pattern of alternating apoptosis and necrosis of acinar cells [52,53]. It is worth noting that smoking and alcohol have a synergistic effect at multiple levels [54]. Nicotine not only enhances the oxidative stress response through the α7 nicotinic acetylcholine receptor, but its metabolite 4 – (methylnitrosamino) – 1 – (3 – pyridyl) – 1 – butanone (NNK) can directly induce DNA damage in pancreatic duct cells [55]. This dual exposure doubles the risk of pancreatitis in smokers compared to those who only consume alcohol. This synergistic effect is particularly significant in Southeast Asia, where the smoking rate is high worldwide [56].

This study represents the inaugural effort to holistically evaluate the global burden of pancreatitis induced by alcohol consumption leveraging the GBD 2021 dataset., providing important insights into an preventable yet under – prioritized

**Table 6. The case number and ASR of Deaths of pancreatitis due to alcohol use in 1990 and 2021 for female by SDI quintiles and by GBD regions, with EAPC from 1990 to 2021.**

| | 1990 | | 2021 | | EAPC (95%CI) 1990–2021 |
|---|---|---|---|---|---|
| | Number (95%UIs) | ASR (95%UIs) | Number (95%UIs) | ASR (95%UIs) | |
| Global | 1527.7 (898.5-2274.54) | 0.07 (0.04-0.11) | 2185.58 (1177.79-3277.93) | 0.05 (0.03-0.07) | −1.41 (−1.66 to −1.17) |
| SDI quintiles | | | | | |
| High SDI | 714.48 (412.93-1059.15) | 0.11 (0.07-0.17) | 894.88 (478.7-1339.75) | 0.08 (0.04-0.11) | −1.26 (−1.36 to −1.16) |
| High-middle SDI | 650.01 (370.12-962.57) | 0.12 (0.07-0.18) | 816.74 (424.16-1251.68) | 0.08 (0.04-0.12) | −1.36 (−1.84 to −0.88) |
| Middle SDI | 101.62 (42.56-174.81) | 0.02 (0.01-0.03) | 265.21 (128.47-421.8) | 0.02 (0.01-0.03) | 0.42 (0.28 to 0.56) |
| Low-middle SDI | 36.67 (12.78-68.13) | 0.01 (0-0.02) | 137.52 (63.48-235.86) | 0.02 (0.01-0.03) | 1.83 (1.65 to 2) |
| Low SDI | 22.13 (8.21-43.11) | 0.02 (0.01-0.04) | 68.19 (30.31-116.97) | 0.02 (0.01-0.04) | 1.01 (0.92 to 1.1) |
| GBD regions | | | | | |
| Andean Latin America | 8.93 (0.49-19.56) | 0.07 (0.01-0.15) | 20.12 (4.34-40.6) | 0.06 (0.01-0.13) | 0.11 (−0.31 to 0.53) |
| Australasia | 15.29 (8.81-23.61) | 0.12 (0.07-0.18) | 24.38 (12.79-39.17) | 0.08 (0.04-0.12) | −0.96 (−1.16 to −0.76) |
| Caribbean | 4.09 (1.36-7.31) | 0.03 (0.01-0.05) | 7.33 (2.35-12.9) | 0.03 (0.01-0.05) | 0.14 (0.03 to 0.26) |
| Central Asia | 25.13 (9.11-44.98) | 0.09 (0.03-0.16) | 23.84 (9.27-41.88) | 0.05 (0.02-0.09) | −3 (−3.41 to −2.59) |
| Central Europe | 133.06 (59.59-216) | 0.16 (0.07-0.27) | 152.19 (71.98-250.17) | 0.13 (0.06-0.21) | −0.87 (−1.09 to −0.65) |
| Central Latin America | 15.1 (4.3-28.2) | 0.03 (0.01-0.05) | 47.59 (16.89-87.3) | 0.03 (0.01-0.06) | 0.56 (0.34 to 0.78) |
| Central Sub-Saharan Africa | 0.75 (0.13-1.9) | 0.01 (0-0.02) | 2.57 (0.89-5.72) | 0.01 (0-0.02) | 1.75 (1.07 to 2.44) |
| East Asia | 60.98 (21.03-118.16) | 0.01 (0-0.03) | 94.7 (37.45-168.76) | 0.01 (0-0.02) | −1.51 (−1.82 to −1.2) |
| Eastern Europe | 251.95 (134.65-394.68) | 0.16 (0.09-0.26) | 427.32 (214.98-665.83) | 0.26 (0.14-0.4) | 1.25 (0.29 to 2.23) |
| Eastern Sub-Saharan Africa | 3.32 (1.18-6.8) | 0.01 (0-0.02) | 10.45 (4.14-20.5) | 0.01 (0-0.02) | 1.04 (0.94 to 1.13) |
| High-income Asia Pacific | 71.98 (38.9-111.87) | 0.07 (0.04-0.1) | 79.62 (35.15-128.35) | 0.03 (0.01-0.04) | −3.84 (−4.08 to −3.59) |
| High-income North Pacific | 128.35 (82.71-214.85) | 0.07 (0.04-0.11) | 292.58 (151.43-446.66) | 0.09 (0.05-0.13) | −3.84 (−4.08 to −3.59) |
| North Africa and Middle East | 2.35 (0.82-4.68) | 0 (0-0.01) | 3.58 (1.19-6.69) | 0 (0−0) | −1.76 (−2.02 to −1.51) |
| Oceania | 0.04 (0.01-0.1) | 0 (0−0) | 0.06 (0.01-0.14) | 0 (0−0) | −1.37 (−1.75 to −0.98) |
| South Asia | 16.06 (4.02-37.38) | 0 (0-0.01) | 67.59 (27.08-129.76) | 0.01 (0-0.02) | 2.59 (2.19 to 3) |
| Southeast Asia | 8.49 (3.4-16.59) | 0.01 (0-0.01) | 25.96 (12.92-45.7) | 0.01 (0-0.01) | 0.37 (0.23 to 0.5) |
| Southern Latin America | 83.57 (49.53-121.3) | 0.33 (0.2-0.48) | 68.02 (33.86-109.6) | 0.14 (0.07-0.23) | −2.13 (−2.5 to −1.75) |

*(Continued)*

**Table 6.** (Continued)

| | 1990 | | 2021 | | EAPC (95%CI) 1990–2021 |
|---|---|---|---|---|---|
| | Number (95%UIs) | ASR (95%UIs) | Number (95%UIs) | ASR (95%UIs) | |
| Southern Sub-Saharan Africa | 5.06 (2.58-8.18) | 0.03 (0.01-0.05) | 10.59 (4.98-16.63) | 0.03 (0.01-0.05) | 0.22 (0.04 to 0.4) |
| Tropical Latin America | 34.32 (17.64-56.39) | 0.06 (0.03-0.1) | 115.83 (55.99-186.74) | 0.08 (0.03-0.13) | 1.19 (0.78 to 1.6) |
| Western Europe | 614.75 (360.57-870.4) | 0.18 (0.11-0.25) | 594 (313.47-894.41) | 0.1 (0.06-0.15) | −1.89 (−1.98 to −1.8) |
| Western Sub-Saharan Africa | 34.36 (12.52-68.69) | 0.08 (0.03-0.15) | 117.26 (51.92-198.51) | 0.11 (0.05-0.18) | 1.08 (0.98 to 1.17) |

**Abbreviations:** ASR, age-standardized rate; DALYs, disability-adjusted life-years;SDI, sociodemographic index; GBD, Global Burden of Diseases; EAPC, estimated annual percentage change; CI, confidence interval.

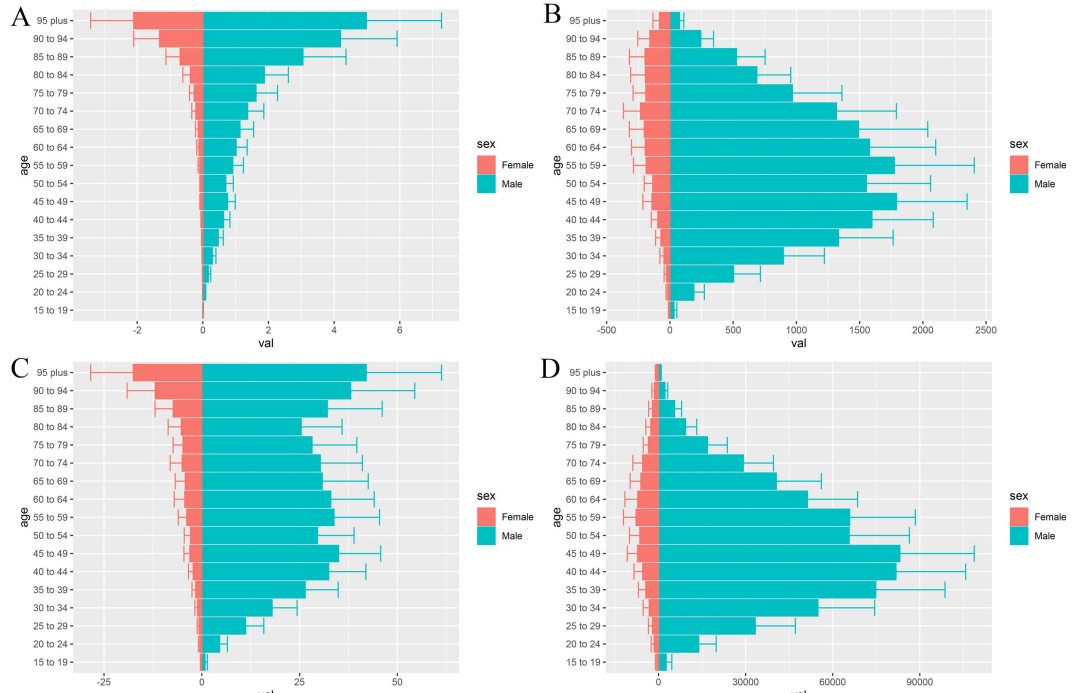

**Fig 1. (A) Age and sex distribution of ASMR; (B) Age and sex distribution of all deaths numbers;(C)Age and sex distribution of ASDR;(D)Age and sex distribution of all DALY numbers. Abbreviations:** ASMR, age-standardized mortality rate; ASDR,age-standardized disability-adjusted life year.

public health challenge. By quantifying the DALYs and mortality rates related to alcohol – associated pancreatitis in various regions, our findings highlight the urgent need for targeted interventions in populations with a high burden, especially in countries and regions where differences are concentrated in alcohol consumption patterns and healthcare accessibility. These results emphasize the importance of incorporating alcohol control measures, such as taxation, supply restrictions, and public awareness campaigns, into national non – communicable disease prevention strategies, as outlined in the World Health Organization (WHO)'s Global Alcohol Action Plan [57].

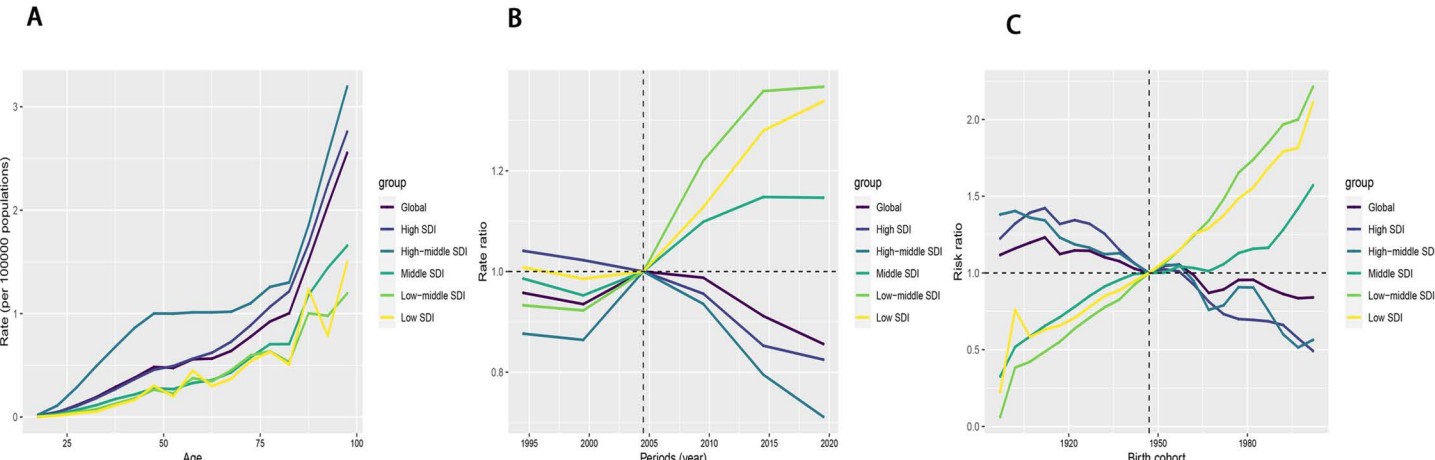

**Fig 2. The APC analysis of the ASDR of pancreatitis due to alcohol use in global and SDI regions. Abbreviations:** APC, Age-period-cohort; ASDR, age-standardized disability-adjusted life year.

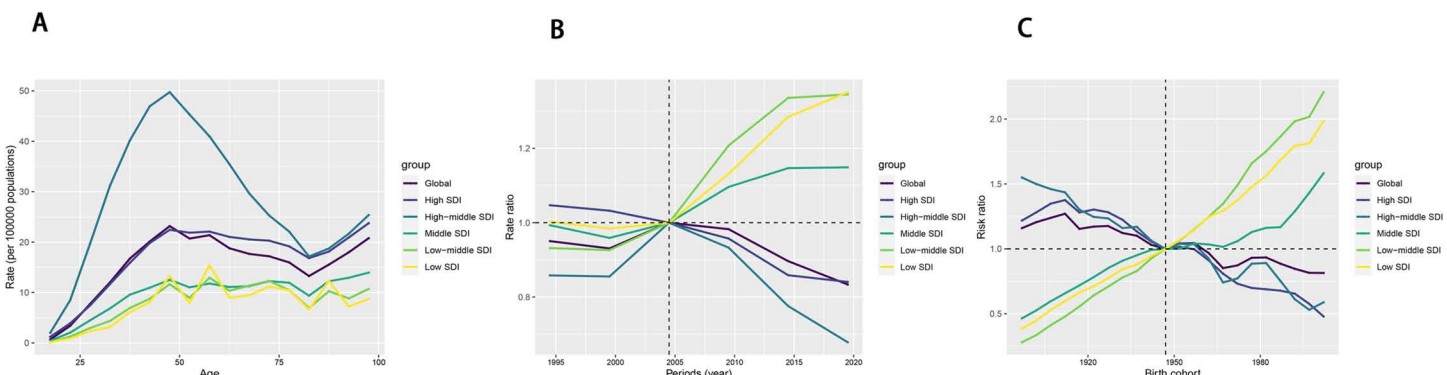

**Fig 3. The APC analysis of the ASMR of pancreatitis due to alcohol use in global and SDI regions. Abbreviations:** APC, Age-period-cohort; ASMR, age-standardized mortality rate.

The predicted inflection point in ASMR after 2034 may be driven by multiple dynamic factors. Firstly, the current elderly population experienced the peak of global alcohol consumption during their prime years (1990–2010), and alcohol-related pancreatitis typically has a latency period of 20–30 years, leading to a concentrated release of disease risk among this group as they enter old age. Secondly, although medical advancements (such as intensive care management) have reduced the early mortality rate of acute pancreatitis, their efficacy in treating late-stage complications of chronic pancreatitis is limited, and the cumulative organ damage among survivors will dominate mortality changes in the next decade. Additionally, the impact of the expansion of emerging alcohol markets in low- and middle-income regions has a lag effect, and the disease burden will further manifest after 2040. This inflection point suggests that priority should be given to intervening in the historically highly exposed elderly population and controlling the accessibility of alcohol in emerging markets.

Furthermore, the observed heterogeneity in the burden between genders and age groups underscores the need for tailored approaches. For example, early screening and brief interventions for middle – aged and elderly men in high – risk areas should be prioritized, while also addressing the sociocultural norms that normalize excessive alcohol consumption. In regions with relatively underdeveloped medical services, enhancing the diagnostic capabilities for the causes of

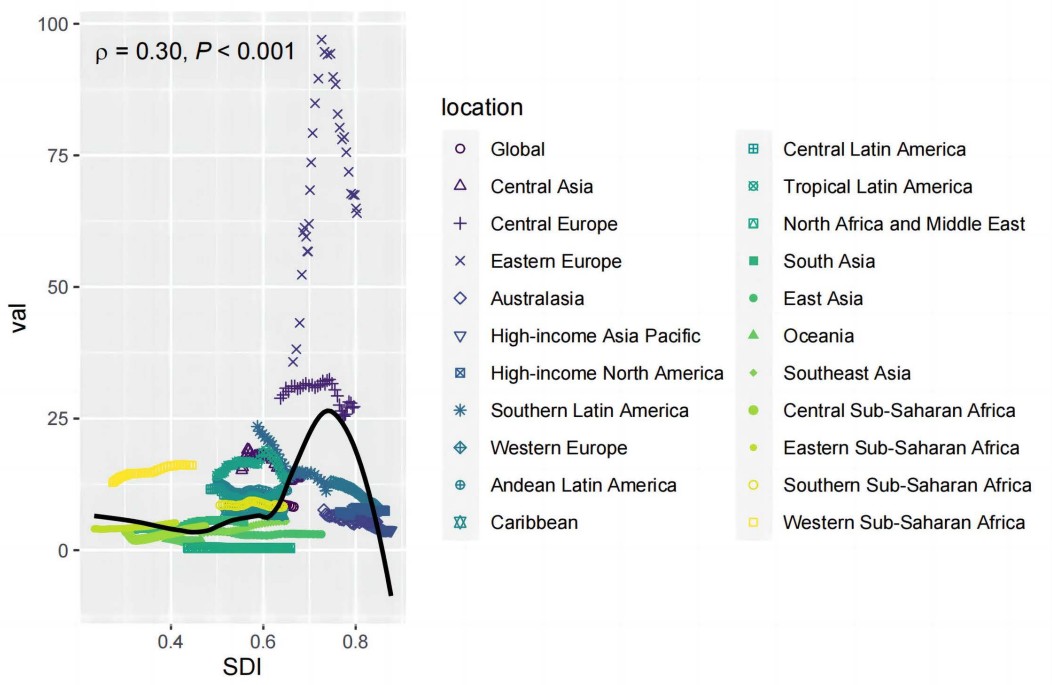

**Fig 4. Relationship between ASDR of pancreatitis due to alcohol use and SDI. Abbreviations:** ASDR,age-standardized disability-adjusted life year;SDI,sociodemographic index.

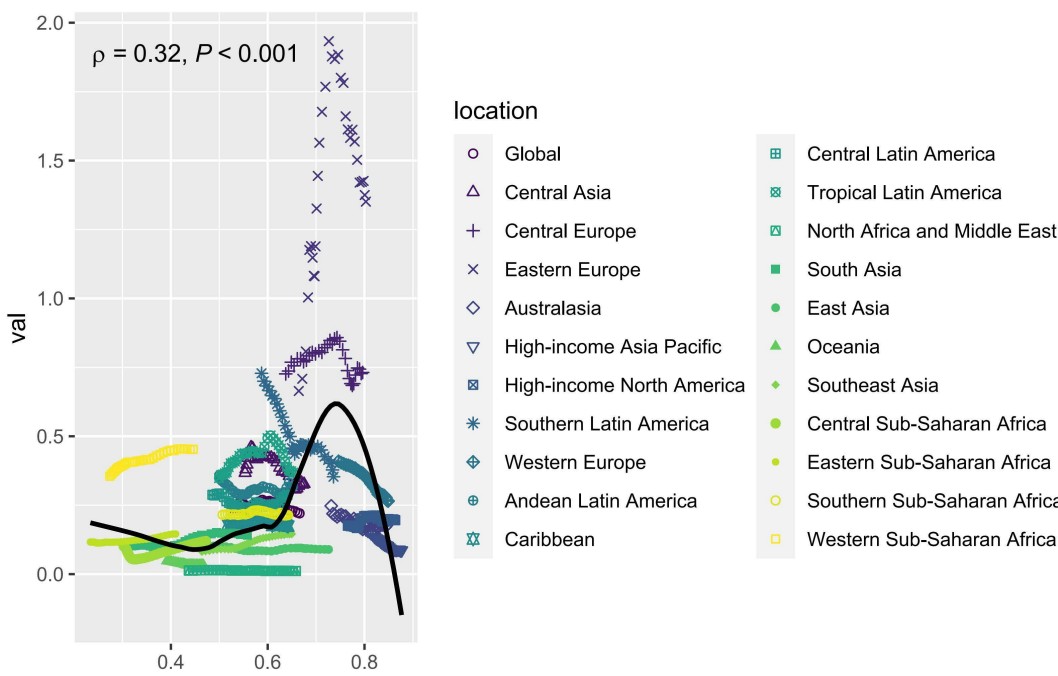

**Fig 5. Relationship between ASMR of pancreatitis due to alcohol use and SDI. Abbreviations:** ASMR, age-standardized mortality rate;SDI,socio-demographic index.

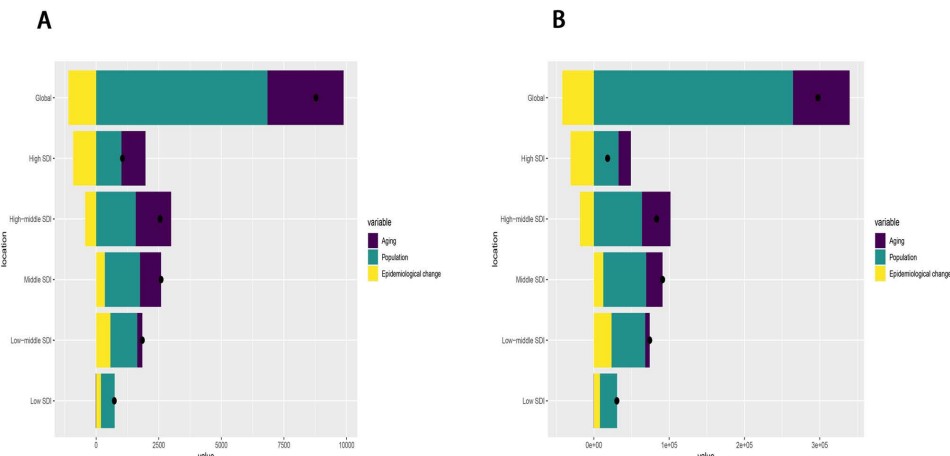

**Fig 6. Changes in (A) Deaths and (B) DALYs of pancreatitis due to alcohol use according to aging, population growth and epidemiological change from 1990 to 2021 at global level by SDI quintile.** The black dot denotes the overall value of the change resulting from all three components. For each component, the magnitude of a positive value suggests a corresponding increase in pancreatitis due to alcohol use DALYs attributed to the component; the magnitude of a negative value suggests a corresponding decrease in pancreatitis due to alcohol use DALYs attributed to the component. **Abbreviations:** DALYs, disability-adjusted life-years; SDI, sociodemographic index.

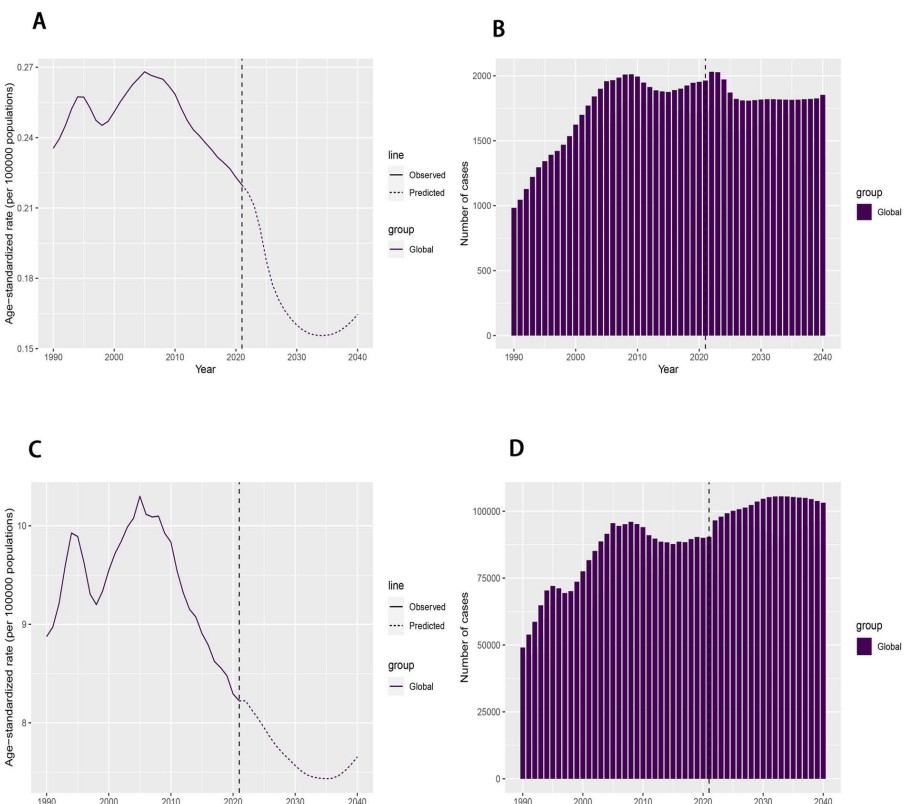

**Fig 7. (A) The rojected ASMR to 2040; (B) The projected number of death cases to 2040; (C) The projected ASDR to 2040; (D)The projected number of DALY cases through 2040;of pancreatitis due to alcohol useglobally. Abbreviations:** ASMR, age-standardized mortality rate; ASDR,age-standardized disability-adjusted life year;DALY, disability-adjusted life-year.

pancreatitis and treatment infrastructure can alleviate long – term complications. Finally, in terms of public health policies, inter – sectoral cooperation among the Ministry of Health, economic policymakers, and community organizations is required to implement cost – effective alcohol control policies, so as to prevent alcoholic pancreatitis at its root.

When interpreting the findings of this study, several inherent limitations should be taken into consideration. First, the GBD 2021 database relies on modeled estimates and aggregated data, which may introduce inherent biases due to underreporting or misclassification of alcohol-related pancreatitis cases. Especially in regions with low social population index where medical facilities and economy are underdeveloped, the reported number of pancreatitis cases may be underestimated due to limited diagnostic means, which may lead to a lower ASMR. Second, the accuracy of alcohol consumption data varies globally; self-reported surveys (a primary source for GBD alcohol metrics) are prone to recall bias and social desirability bias, potentially underestimating true consumption levels. Finally, the diagnostic criteria for pancreatitis and its etiological attribution to alcohol use may differ across countries, leading to heterogeneity in case definitions. We hope that future GBD databases can further improve these existing issues.

## 5 Conclusion

In summary, pancreatitis caused by alcohol use is a major global public health problem, with a significant increase in disease burden between 1990 and 2021, especially among middle-aged and older men. Analysis based on the GBD database shows that changes in the disease burden of alcoholic pancreatitis are primarily driven by two major demographic factors, population growth and population aging, and this trend and our findings predict a likely dramatic increase in the number of cases of alcoholic pancreatitis globally in the coming decades. The findings highlight the dual challenges facing the prevention and control of alcoholic pancreatitis: the continuing increase in the burden of disease and the significant heterogeneity of the burden distribution across different regions of the world. These findings provide an important evidence-based basis for optimizing public health policy making and medical resource allocation. It is recommended that global health policymakers adopt more targeted intervention strategies, implement flexible prevention and control measures, and focus on building personalized health care systems to address the differentiated health needs of different countries.

## Supporting information

**S1 Table. Burden of alcoholic pancreatitis by all countries (Male).**
(XLSX)

**S2 Table. Burden of alcoholic pancreatitis by all countries (Female).**
(XLSX)

**S3 Table. Burden of alcoholic pancreatitis by all countries (both).**
(XLSX)

**S4 Table. EAPC values for the burden of alcoholic pancreatitis in all countries.**
(XLSX)

**S5 Table. Age data of Age-Period-Cohort analysis of alcoholic pancreatitis.**
(XLSX)

**S6 Table. Period data of Age-Period-Cohort analysis of alcoholic pancreatitis.**
(XLSX)

**S7 Table. Cohort data of Age-Period-Cohort analysis of alcoholic pancreatitis.**
(XLSX)

**S8 Table. Results of decomposition analysis of alcoholic pancreatitis globally and in the five SDI regions.**
(XLSX)

**S9 Table. BAPC prediction of the global burden of alcoholic pancreatitis.**
(XLSX)

## Acknowledgments

Thanks to the IHME and the Global Burden of Disease study collaborations.

## Author contributions

**Conceptualization:** Letai Li, Yuxiang Luo.

**Data curation:** Letai Li.

**Formal analysis:** Letai Li, Haibing Xiong.

**Methodology:** Jiajie Leng.

**Project administration:** Zhenrui Cao.

**Software:** Yaowen Zhang.

**Supervision:** Yang Lei, Rui Tao, Yingjiu Jiang.

**Visualization:** Siyu Li.

**Writing – original draft:** Letai Li, Yaowen Zhang, Jiajie Leng.

**Writing – review & editing:** Jiajie Leng, Zhongjun Wu, Rui Tao, Yingjiu Jiang.

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
