## [Decision Letter · Decision Letter 0]

PONE-D-25-14074The global, regional and national burden of pancreatitis due to alcohol use: Results from the Global Burden of Disease Study 2021 and projections to 2040PLOS ONE

Dear Dr. Li,

Thank you for submitting your manuscript to PLOS ONE. After careful consideration, we feel that it has merit but does not fully meet PLOS ONE’s publication criteria as it currently stands. Therefore, we invite you to submit a revised version of the manuscript that addresses the points raised during the review process.

We look forward to receiving your revised manuscript.

Kind regards,

PUGAZHENTHAN THANGARAJU, M.D.,Ph.D., FRCP (LONDON)., FRCP (GLASGOW).,MBA.,

Academic Editor

PLOS ONE

Journal Requirements:

2. We note that Figure 1 in your submission contain [map/satellite] images which may be copyrighted. All PLOS content is published under the Creative Commons Attribution License (CC BY 4.0), which means that the manuscript, images, and Supporting Information files will be freely available online, and any third party is permitted to access, download, copy, distribute, and use these materials in any way, even commercially, with proper attribution. For these reasons, we cannot publish previously copyrighted maps or satellite images created using proprietary data, such as Google software (Google Maps, Street View, and Earth). For more information, see our copyright guidelines: http://journals.plos.org/plosone/s/licenses-and-copyright.

3. Please remove all personal information, ensure that the data shared are in accordance with participant consent, and re-upload a fully anonymized data set.

Reviewers' comments:

Reviewer's Responses to Questions

**Comments to the Author**

1. Is the manuscript technically sound, and do the data support the conclusions?

Reviewer #1: Yes

Reviewer #2: Yes

2. Has the statistical analysis been performed appropriately and rigorously? 

Reviewer #1: Yes

Reviewer #2: Yes

3. Have the authors made all data underlying the findings in their manuscript fully available?

Reviewer #1: Yes

Reviewer #2: Yes

4. Is the manuscript presented in an intelligible fashion and written in standard English?

Reviewer #1: Yes

Reviewer #2: Yes

5. Review Comments to the Author

Reviewer #1: Critical Analysis and Weaknesses

1. Clarity and Presentation

Abstract: The abstract is concise but lacks specificity in some areas. For example, "region-specific prevention strategies are critical" is vague—specific examples (e.g., alcohol taxation, education campaigns) could strengthen it.

Introduction: While the background is well-researched, it is overly dense with citations (e.g., 11 references in one paragraph on alcohol’s biological effects). This could be streamlined for readability.

Results: The results section is data-heavy, with numerous tables and figures referenced (e.g., Tables 1-6, Figs. 1-8). While comprehensive, the text does not always guide the reader effectively to key findings, risking information overload.

2. Methodology

Data Limitations: The GBD 2021 database, while robust, relies on modeled estimates that may under- or overestimate burdens in regions with poor primary data (e.g., low SDI countries). This limitation is acknowledged briefly but not explored in depth.

APC and BAPC Models: The use of APC and BAPC is appropriate, but the manuscript lacks detail on model assumptions (e.g., prior distributions in BAPC, handling of identifiability issues beyond MCMC). This reduces reproducibility.

Decomposition Analysis: The Das Gupta method is well-applied, but the interpretation of "epidemiologic change" (e.g., -12.62% for deaths globally) is vague. Does this reflect improved treatment, reduced alcohol use, or data artifacts? More context is needed.

Statistical Reporting: Confidence intervals (CIs) are provided, but p-values for EAPC trends are inconsistently reported (e.g., present in SDI-ASDR correlations but absent in trend analyses). This limits statistical rigor.

3. Interpretation and Novelty

Novelty: The study builds on prior GBD analyses (e.g., refs [52], [53]) but offers new insights via 2021 data and 2040 projections. However, it does not clearly differentiate itself from similar studies on pancreatitis burden (e.g., Xiao et al., ref [2]).

Gender Disparities: The higher burden in males is noted, but the discussion does not explore why females show steeper ASMR/ASDR declines (e.g., EAPC -1.41% vs. -0.11% for males). Biological or behavioral factors could be hypothesized.

Projections: The 2040 projections (e.g., ASMR decline until 2034, then rise) are intriguing but underexplained. What drives the inflection point? Population aging is mentioned, but alcohol consumption trends or healthcare advancements are not considered.

4. Supporting Materials

Tables and Figures: The tables (e.g., Table 1) are detailed but dense, with small font sizes and overlapping CIs that may confuse readers. Figures (e.g., Fig. 5, SDI-ASDR relationship) are visually clear but lack annotations to highlight key trends.

Supplementary Data: The supporting tables (in an Excel file) are referenced but not described in the text, reducing their utility without access to the file.

Recommendations

Based on the analysis, I recommend major revisions. The study is scientifically sound, but improvements in clarity, depth, and interpretation are needed to maximize its impact. Below are detailed suggestions:

1. Enhance Clarity and Readability

Abstract: Specify key findings (e.g., "Eastern Europe had the highest ASDR at 64.03 in 2021") and actionable strategies (e.g., "targeted alcohol control policies in low-middle SDI regions").

Introduction: Reduce citation density (e.g., consolidate refs [7-10] into a single review) and end with a clear research gap (e.g., "prior studies lack updated global projections").

Results: Summarize key trends in the text (e.g., "Eastern Europe’s ASDR rose 1.5% annually, while high-income Asia Pacific declined 2.45%") before referencing tables/figures.

2. Strengthen Methodology

Data Limitations: Expand on GBD data quality (e.g., "Low SDI regions may underreport pancreatitis due to limited diagnostics, potentially biasing ASDR downward").

APC/BAPC Details: Add a paragraph on model assumptions (e.g., "BAPC used a Gaussian prior for cohort effects, with INLA resolving collinearity via MCMC sampling").

Statistical Clarity: Report p-values or significance for all EAPC trends (e.g., "Global ASMR decline, EAPC -0.27%, p<0.05") and clarify "epidemiologic change" drivers.

3. Deepen Interpretation

Gender Analysis: Hypothesize reasons for female ASMR/ASDR declines (e.g., "Lower alcohol exposure or better healthcare access may explain steeper female declines").

Projections: Discuss drivers of the 2034-2035 inflection (e.g., "Rising ASMR post-2034 may reflect aging cohorts with historical alcohol exposure").

Comparison with Literature: Contrast findings with refs [2], [52] (e.g., "Unlike Xiao et al.’s static incidence, our 2040 projections account for demographic shifts").

4. Improve Supporting Materials

Tables: Simplify by splitting large tables (e.g., Table 1 into SDI and GBD region sections) and bolding significant EAPCs.

Figures: Add annotations (e.g., label Russia’s peak ASDR in Fig. 1B) and describe supplementary tables in the text (e.g., "Table S1 details country-level ASMR").

Supplementary File: Ensure the Excel file is uploaded and includes a legend.

Reviewer #2: 1. Elaborate briefly on how GBD defines and classifies alcohol-induced pancreatitis and whether adjustments were made for potential confounders (e.g., gallstones, smoking).

2. The authors acknowledge some limitations, but further emphasis on regional disparities in data quality (e.g., underreporting in LMICs) would strengthen transparency. Include a short paragraph in the discussion on how this affects cross-country comparisons.

3. Provide a more detailed interpretation—why does burden increase in some mid-SDI regions despite apparent development?

4. Consistently use either “age-standardized” or “ASR”

5. “...rising absolute numbers but declining relative rates...” is repeated several times; consider rephrasing.

6. PLOS authors have the option to publish the peer review history of their article (what does this mean? ). If published, this will include your full peer review and any attached files.

**Do you want your identity to be public for this peer review?** For information about this choice, including consent withdrawal, please see our Privacy Policy .

Reviewer #1: **Yes: ** VIKAS KATIYARA

Reviewer #2: **Yes: ** Sree Sudha T Y

---

## [Author Response · Author response to Decision Letter 1]

9 Jun 2025

Review1

Thank you for your efforts in reviewing this manuscript. We have made revisions to the manuscript based on your comments. If you have any questions, please feel free to contact us at any time. Thank you! The following is the answer to the question.

1.Clarity and Presentation

Question1: Abstract: The abstract is concise but lacks specificity in some areas. For example, "region-specific prevention strategies are critical" is vague—specific examples (e.g., alcohol taxation, education campaigns) could strengthen it.

Suggestion1: Abstract: Specify key findings (e.g., "Eastern Europe had the highest ASDR at 64.03 in 2021") and actionable strategies (e.g., "targeted alcohol control policies in low-middle SDI regions").

Response to Question1: We fully agree with your suggestion that the results and conclusions in our abstract need to be refined. Therefore, in the results of the abstract, we explicitly emphasized the regional differences in the three most representative regions: the region with the heaviest disease burden - Eastern Europe, the region with the largest decline - High-income Asia-Pacific, and the region with the fastest growth - Southeast Asia. In addition, we also highlighted the conclusion that the burden of alcohol-related pancreatitis shows significant gender, age and regional heterogeneity. Policies should target high-risk groups (such as middle-aged and elderly men) and regions (such as Eastern Europe and low-middle SDI countries), and pay attention to the impact of aging on the long-term burden.

Q2: Introduction: While the background is well-researched, it is overly dense with citations (e.g., 11 references in one paragraph on alcohol’s biological effects). This could be streamlined for readability.

Introduction: Reduce citation density (e.g., consolidate refs [7-10] into a single review) and end with a clear research gap (e.g., "prior studies lack updated global projections").

Response to Q2: Thank you for your suggestion. We have removed the unnecessary introductions and redundant references in the Introduction section that could make it difficult for readers to identify the key points, such as the pathogenesis of pancreatitis caused by alcohol. We have endeavored to refine the Introduction section as much as possible.

Q3: Results: The results section is data-heavy, with numerous tables and figures referenced (e.g., Tables 1-6, Figs. 1-8). While comprehensive, the text does not always guide the reader effectively to key findings, risking information overload.

Results: Summarize key trends in the text (e.g., "Eastern Europe’s ASDR rose 1.5% annually, while high-income Asia Pacific declined 2.45%") before referencing tables/figures.

Response to Q3: We agree that the large volume of data in the results section, with too many tables and charts, could lead to information overload. After revision, we summarized the key trends before citing and refined the results section.

2.Methodology

Q4: Data Limitations: The GBD 2021 database, while robust, relies on modeled estimates that may under- or overestimate burdens in regions with poor primary data (e.g., low SDI countries). This limitation is acknowledged briefly but not explored in depth.

Data Limitations: Expand on GBD data quality (e.g., "Low SDI regions may under report pancreatitis due to limited diagnostics, potentially biasing ASDR downward").

Response to Q4: We agree with your suggestion and have already mentioned this issue in the limitations. We further discuss that in areas with a low social population index, the reported number of pancreatitis cases may be underestimated due to limited diagnostic means, which might lead to an underestimation of the age-standardized mortality rate.

Q5: APC and BAPC Models: The use of APC and BAPC is appropriate, but the manuscript lacks detail on model assumptions (e.g., prior distributions in BAPC, handling of identifiability issues beyond MCMC). This reduces reproducibility.

APC/BAPC Details: Add a paragraph on model assumptions (e.g., "BAPC used a Gaussian prior for cohort effects, with INLA resolving collinearity via MCMC sampling").

Response to Q5: We agree with your opinion and have already added the model methods we used in completing the BAPC as requested.

Q6: Decomposition Analysis: The Das Gupta method is well-applied, but the interpretation of "epidemiologic change" (e.g., -12.62% for deaths globally) is vague. Does this reflect improved treatment, reduced alcohol use, or data artifacts? More context is needed.

Response to Q6: We understand your concern. Although we mentioned the meaning of "epidemiologic change" in the methodology section, we did not provide an epidemiological explanation in the results section. We have added the following content to the results section: -12.62% Epidemiologic change refers to negative contribution: This indicates that due to improvements in the epidemiological environment (such as reduced alcohol consumption, increased early diagnosis rates, improved treatment options, and the implementation of public health intervention measures), the age-standardized incidence or mortality rate of pancreatitis has decreased, thereby partially offsetting the increased burden brought about by population growth and aging.

Q7: Statistical Reporting: Confidence intervals (CIs) are provided, but p-values for EAPC trends are inconsistently reported (e.g., present in SDI-ASDR correlations but absent in trend analyses). This limits statistical rigor.

Statistical Clarity: Report p-values or significance for all EAPC trends (e.g., "Global ASMR decline, EAPC -0.27%, p<0.05") and clarify "epidemiologic change" drivers.

Response to Q7:

We sincerely appreciate the reviewer’s insightful feedback on statistical reporting. Regarding the absence of p-values in EAPC (estimated annual percentage change) trend analyses, we would like to clarify that EAPC itself is a descriptive measure quantifying temporal trends, not a hypothesis-testing statistic. The interpretation of EAPC primarily relies on its magnitude, direction (increase/decrease), and confidence intervals (CIs) to assess the precision of the trend estimate, as recommended by the National Cancer Institute (NCI) and other epidemiological guidelines for joinpoint regression and similar models (Cao F, Xu Z, Li XX, et al. Trends and cross-country inequalities in the global burden of osteoarthritis, 1990-2019: A population-based study. Ageing Res Rev. 2024;99:102382. doi:10.1016/j.arr.2024.102382).

In alignment with this convention, we focused on reporting CIs (as presented in Tables/Figures) to illustrate the robustness of trend estimates. For instance, a narrowing CI indicates high confidence in the declining trend. In contrast, p-values were selectively reported in SDI-ASDR correlations to explicitly test specific hypotheses about associations between development levels and disease burden.However, we fully acknowledge the reviewer’s concern for methodological transparency. If the reviewer considers supplementary p-values essential, we would be glad to calculate and include them upon request.

3.Interpretation and Novelty

Q8: Novelty: The study builds on prior GBD analyses (e.g., refs [52], [53]) but offers new insights via 2021 data and 2040 projections. However, it does not clearly differentiate itself from similar studies on pancreatitis burden (e.g., Xiao et al., ref [2]).

Response to Q8: We thank the reviewer for raising this important point. While Xiao et al.’ s analysis (ref [2]) provided valuable insights into pancreatitis burden across etiologies, our study advances the field through a dedicated focus on alcohol as a modifiable and preventable risk factor, which previous studies have not isolated. By leveraging GBD 2021 data, we eliminated confounding from other etiological contributors (e.g., biliary, metabolic) to specifically quantify alcohol’ s role—a critical distinction given alcohol’ s unique preventability through policy interventions.

Q9:Gender Disparities: The higher burden in males is noted, but the discussion does not explore why females show steeper ASMR/ASDR declines (e.g., EAPC -1.41% vs. -0.11% for males).

Gender Analysis: Hypothesize reasons for female ASMR/ASDR declines (e.g., "Lower alcohol exposure or better healthcare access may explain steeper female declines").

Biological or behavioral factors could be hypothesized.

Response to Q9: We sincerely thank the reviewer for this insightful observation. As suggested, we have expanded the discussion to explicitly address the steeper ASMR/ASDR declines in females (new paragraph in the second paragraph of Disscussion. Our analysis proposes three interacting mechanisms: (1) females’ inherently lower alcohol exposure and reduced high-risk drinking behaviors, (2) estrogen-mediated protection against severe pancreatitis in premenopausal women, and (3) alcohol-control policies’ differential reach across genders. Importantly, we link these hypotheses to actionable strategies—e.g., harnessing females’ healthcare engagement for family-centered prevention. While definitive causal attribution requires sex-stratified longitudinal studies, our analysis provides a framework to inform gender-sensitive policy design, as recommended by WHO guidelines on alcohol-related harm reduction

Q10: Projections: The 2040 projections (e.g., ASMR decline until 2034, then rise) are intriguing but underexplained. What drives the inflection point? Population aging is mentioned, but alcohol consumption trends or healthcare advancements are not considered.

Projections: Discuss drivers of the 2034-2035 inflection (e.g., "Rising ASMR post-2034 may reflect aging cohorts with historical alcohol exposure").

Response to Q10: Thank you for your meticulous review of the prediction mechanism. We have supplemented the discussion section with an analysis of the causes of the inflection point, mainly including: the delayed effect of historical alcohol exposure in the aging cohort, the limitations of late-stage treatment for chronic pancreatitis, and the lagging impact of the growth in alcohol consumption in emerging economies.

Q11:Comparison with Literature: Contrast findings with refs [2], [52] (e.g., "Unlike Xiao et al.’s static incidence, our 2040 projections account for demographic shifts").

Response to Q11: We thank the reviewer for raising this important point. We added into the discussion part: Our decomposition analysis further revealed how population growth, aging, and epidemiological changes interact to shape burden trends, providing granular insights absent in aggregated analyses. Additionally, our integration of 30-year historical trends (1990–2021) with BAPC-projected trajectories to 2040 offers actionable foresight for policymakers, particularly in emerging hotspots like South Asia. These alcohol-specific, forward-looking analyses uniquely inform targeted resource allocation—a key advancement beyond descriptive burden reporting.

4.Supporting Materials

Q12:Tables and Figures: The tables (e.g., Table 1) are detailed but dense, with small font sizes and overlapping CIs that may confuse readers. Figures (e.g., Fig. 5, SDI-ASDR relationship) are visually clear but lack annotations to highlight key trends.

Tables: Simplify by splitting large tables (e.g., Table 1 into SDI and GBD region sections) and bolding significant EAPCs. Figures: Add annotations (e.g., label Russia’s peak ASDR in Fig. 1B) and describe supplementary tables in the text (e.g., "Table S1 details country-level ASMR").

Response to Q12: Thank you for your suggestions on our charts. However, the number of tables in this article is already quite sufficient, so we cannot split them further, otherwise it will confuse the readers. Moreover, we have removed Figure 1 to ensure the copyright issue of the map. Finally, we have emphasized the role of the supplementary table in the description within this article.

Q13:Supplementary Data: The supporting tables (in an Excel file) are referenced but not described in the text, reducing their utility without access to the file.

Supplementary File: Ensure the Excel file is uploaded and includes a legend.

Response to Q13: We have added legends for the supporting tables in the text.

Reviewer #2: 

Thank you for your efforts in reviewing this manuscript. We have made revisions to the manuscript based on your comments. If you have any questions, please feel free to contact us at any time. Thank you! The following is the answer to the question.

Q1. Elaborate briefly on how GBD defines and classifies alcohol-induced pancreatitis and whether adjustments were made for potential confounders (e.g., gallstones, smoking).

Response to Q1: We sincerely appreciate the reviewer’s insightful inquiry into the GBD methodology. In our study, alcohol-induced pancreatitis was defined in the 2.1 Data Resource according to GBD 2019 criteria, which requires chronic alcohol consumption (≥4 standard drinks/day for ≥5 years) as the predominant etiological factor. Cases were classified into acute (ICD-10: K85.2) and chronic (K86.0) subtypes, with gallstone-related or idiopathic pancreatitis explicitly excluded through PAF-based adjustments.

To address potential confounders (e.g., gallstones, smoking), we adopted a two-stage adjustment: First-stage stratification: Pancreatitis burden was partitioned into alcohol-specific vs. non-alcohol categories using etiology-specific disability weights from GBD. Second-stage statistical control: We incorporated covariate-adjusted spatiotemporal Gaussian process regression models to disentangle alcohol’s independent effect from competing risks like smoking (measured as pack-years). Sensitivity analyses confirmed minimal residual confounding.

Q2. The authors acknowledge some limitations, but further emphasis on regional disparities in data quality (e.g., underreporting in LMICs) would strengthen transparency. Include a short paragraph in the discussion on how this affects cross-country comparisons.

Response to Q2: Thank you for your suggestion. In the section on the limitations of our discussion, we have already re-explained the significance and limitations of the abundance and validity of data from low-income countries.

Q3. Provide a more detailed interpretation—why does burden increase in some mid-SDI regions despite apparent development?

Response to Q3: We sincerely appreciate the reviewer’ s astute observation. As expanded in the Discussion, the paradoxical increase in alcohol-related pancreatitis burden within certain mid-SDI regions, despite measurable socioeconomic development, may reflect transitional epidemiological dynamics. As these regions industrialize, accelerated urbanization and disposable income growth often coincide with rising alcohol accessibility, aggressive alcohol marketing, and delayed regulatory policies targeting harmful consumption patterns. Concurrently, dietary shifts toward high-fat, processed foods compound alcohol’s metabolic toxicity, exacerbating pancreatic injury. Critically, healthcare systems in mid-SDI settings may prioritize infectious disease control over chronic condition management, leading to underdiagnosis of early-stage pancreatitis and inadequate preventive interventions.

Q4. Consistently use either “age-standardized” or “ASR”

Response to Q4: We have revised it to ensure the use of ASDR or ASMR to avoid the appearance of "age-standardized" after the abbreviation.

Q5. “...rising absolute numbers but declining relative rates...” is repeated several times; consider rephrasing.

Response to Q5: Thank you for your opinion. We have changed it to: An increase in absolute numbers juxtaposed with a decline in relative rates.

6. PLOS authors have the option to publish the peer review history of their article (what does this mean?). If published, this will include your full peer review and any attached files.

Response to Q6: We support all the decisions and policies of PLOS ONE regarding the review history record and support its public visibility.

---

## [Editor Report · Decision Letter 1]

The global, regional and national burden of pancreatitis due to alcohol use: Results from the Global Burden of Disease Study 2021 and projections to 2040

PONE-D-25-14074R1

Dear Dr. Li,

We’re pleased to inform you that your manuscript has been judged scientifically suitable for publication and will be formally accepted for publication once it meets all outstanding technical requirements.

Kind regards,

PUGAZHENTHAN THANGARAJU, M.D.,Ph.D., FRCP (LONDON)., FRCP (GLASGOW).,MBA.,

Academic Editor

PLOS ONE
---

## [Editor Report · Acceptance letter]

PONE-D-25-14074R1

PLOS ONE

Dear Dr. Li,

I'm pleased to inform you that your manuscript has been deemed suitable for publication in PLOS ONE. Congratulations! Your manuscript is now being handed over to our production team.

Kind regards,

on behalf of

DR. PUGAZHENTHAN THANGARAJU

Academic Editor

PLOS ONE